# INFORMATION-THEORETIC GENERALIZATION ANALYSIS FOR VECTOR-QUANTIZED VAES

## ABSTRACT

Encoder–decoder models, which transform input data into latent variables, have achieved a significant success in machine learning. Although the generalization capability of these models has been theoretically analyzed in supervised learning focusing on the complexity of latent variables, the contribution of latent variables in generalization and data generation capabilities are less explored theoretically in unsupervised learning. To address this gap, our study leverages information-theoretic generalization error analysis (IT analysis). Using the supersample setting in recent IT analysis, we demonstrate that the generalization gap for reconstruction loss can be evaluated through mutual information related to the posterior distribution of latent variables, conditioned on the input data, without relying on the decoder's information. We also introduce a novel permutation-symmetric supersample setting, which extends the existing IT analysis and shows that regularizing the encoder's capacity leads to generalization. Finally, we guarantee the 2-Wasserstein distance between the true data distribution and the generated data distribution, offering insights into the model's data generation capabilities.

## 1 INTRODUCTION

Encoder–decoder models have achieved a significant success in machine learning (Goodfellow et al., 2016). Typically, the encoder extracts information from input data to generate latent variables, called representations, and the decoder uses these representations to output predictions. In supervised learning, these models are trained by minimizing the empirical loss, and the regularization of latent variables helps prevent overfitting, thereby improving generalization performance. As detailed in Sec 1.1, studies on encoder–decoder models have focused on not only learned parameters but also on the complexity of latent variables, through principles such as the minimum description length (MDL) (Grnwald et al., 2005), PAC-Bayes theory (McAllester, 1998), information-theoretic generalization error analysis (IT analysis) (Xu & Raginsky, 2017), and the information bottleneck (IB) hypothesis (Tishby et al., 2000). These approaches have demonstrated, both numerically and theoretically, that generalization can be characterized by the complexity of latent variables (Sefidgaran et al., 2023).

Encoder–decoder models are also popular in unsupervised learning, particularly in deep generative models. When training these models, we minimize the reconstruction loss, which measures the difference between the original data and the regenerated data obtained by compressing data into latent variables by the encoder and regenerating the data by the decoder. Similarly to supervised learning, the regularization of the latent variables plays a critical role. For example, in the variational autoencoder (VAE) (Kingma, 2013), the objective function corresponds to the lower bound of the log-likelihood. In the case of a Gaussian likelihood, the reconstruction loss corresponds to the squared loss, and the regularization term is the Kullback–Leibler (KL) divergence between the prior and posterior distributions of the latent variables. Despite experimental studies examining this relationship using the IB hypothesis, as noted in Sec 1.1, there is a lack of theoretical analysis focusing on latent variables. Most research has concentrated on encoder and decoder parameters, resulting in a limited understanding of how latent variables contribute to model performance. This study addresses this gap by applying IT analysis to clarify the roles of latent variables in generalization and data generation.

## 1.1 RELATED WORK

The IB hypothesis (Tishby et al., 2000; Shamir et al., 2010) is the most well-known concept for learning meaningful representations in latent variable models. It suggests that representations that retain essential information for prediction while minimizing mutual information (MI) between input data and latent variables lead to better generalization. Such MI in the IB hypothesis has been used as an empirical tool to understand deep learning mechanisms (Tishby & Zaslavsky, 2015; Shwartz-Ziv & Tishby, 2017; Saxe et al., 2019; Goldfeld et al., 2019; Achille & Soatto, 2018a;b). However, debates persist about whether MI alone fully captures generalization. Sefidgaran et al. (2023) provided a detailed discussion of these concerns. For example, Geiger & Koch (2019), Geiger (2021), and Amjad & Geiger (2019) raised both numerical and theoretical concerns, whereas Vera et al. (2018), Hafez-Kolahi et al. (2020), Kawaguchi et al. (2023), and Vera et al. (2023) provided theoretical upper bounds on the generalization error for classification using such MI. However, these studies have several limitations, such as assuming discrete data, the exponential dependence of the bound on MI, and the entropy of input data.

In contrast, in IT analysis (Xu & Raginsky, 2017), the generalization error is evaluated on the basis of the MI between learned parameters and training data. This approach is closely related to the PAC-Bayes theory and has been extended through supersample settings (Steinke & Zakynthinou, 2020) to exploit the symmetry between test and training data. This setting has been applied to the study of generalization based on outputs of functions (Harutyunyan et al., 2021), losses (Hellström & Durisi, 2022; Wang & Mao, 2023), and hypothesis entropy (Dong et al., 2024). The relationship between IT analysis and the IB hypothesis has been discussed from numerical and algorithmic perspectives (Wang et al., 2022; Lyu et al., 2023). More recently, Sefidgaran et al. (2023) theoretically studied latent variable models using IT analysis, demonstrating that generalization can be characterized by the complexity of the encoder and latent variables without relying on decoder information. They also developed a theoretical link among IT analysis, the IB hypothesis, and MDL by using compression bounds (Blum & Langford, 2003).

In unsupervised learning, there have been numerous empirical and qualitative studies to explore model performance using the IB hypothesis and rate-distortion theory (Cover & Thomas, 2012) (Alemi et al., 2018; Blau & Michaeli, 2019; Tschannen et al., 2020; Bond-Taylor et al., 2021), but theoretical advances remain limited. For instance, PAC-Bayes bounds for the reconstruction loss were proposed by Chérief-Abdellatif et al. (2022) on the basis of the PAC-Bayes theory of supervised learning using prior and posterior distributions over the encoder and decoder parameters. Similarly, Epstein & Meir (2019) proposed parameter-based bounds. However, these works fall short of explaining the theoretical role of latent variables. Mbacke et al. (2023) have recently introduced PAC-Bayes bounds that use priors and posteriors over the latent variables, aligning the PAC-Bayes posterior with the variational posterior. Their approach allows for using the posterior distribution, which is conditionally independent between data points, providing guarantees for generalization under the reconstruction loss and data generation capabilities. However, in their work, they assumed fixed encoder and decoder parameters, without considering learning these parameters.

## 1.2 OUR CONTRIBUTIONS

On the basis of existing research findings, we provide a theoretical analysis that guarantees the generalization and data generation capabilities of unsupervised learning models, focusing on latent variables. However, simply extending the analysis of VAEs (Mbacke et al., 2023) results in generalization bounds that depend on learned decoder parameters, obscuring the role of latent variables. Similarly, directly using the IT analysis from supervised learning (Harutyunyan et al., 2021; Hellström & Durisi, 2022) is challenging owing to the difficulty in decoupling the encoder–decoder relationship. Furthermore, the techniques used for classification in latent variable models (Sefidgaran et al., 2023) are insufficient for analyzing the reconstruction loss or data generation, as these require reusing input data and handling conditionally independent posterior distributions.

To address these challenges and advance the theoretical analysis of unsupervised learning with a focus on latent variables, we propose a novel information-theoretic generalization error bound (Theorem 2) for models with finite latent variables, such as vector quantized VAEs (VQ-VAEs) (Van Den Oord et al., 2017) (detailed in Section 2.1). This extends the supersample setting in existing IT analysis (Steinke & Zakynthinou, 2020) incorporating techniques from Mbacke et al. (2023)

and Sefidgaran et al. (2023). Furthermore, we introduce a novel permutation-invariant supersample setting, ensuring that the generalization gap vanishes when the encoder's capacity is appropriately regularized (Theorems 3 and 4). Even without such a constraint, we show that generalization can still be guaranteed if the posterior distribution of latent variables is sufficiently stable (Section 3.3). Finally, we provide a guarantee for the data-generating capability by deriving the upper bound on the 2-Wasserstein distance between the true data distribution and the generated data distribution (Theorem 5). These findings provide the first comprehensive theoretical understanding of how encoders and latent variables contribute to generalization and data generation capabilities.

## 2 PRELIMINARIES

For a random variable (RV) denoted in capital letters, we express its realization with corresponding lowercase letters. Let $p(X)$ denote the distribution of $X$, and let $p(Y|X)$ represent the conditional distribution of $Y$ given $X$. We express the expectation of a random variable $X$ as $\mathbb{E}_{p(X)}$ or $\mathbb{E}_X$. The symbol $I(X;Y)$ represents the MI between $X$ and $Y$, while $I(X;Y|Z)$ is the conditional MI (CMI) between $X$ and $Y$ given $Z$. The Kullback–Leibler (KL) divergence between $p(X)$ and $p(Y)$ is denoted by $\mathrm{KL}(p(X)\|p(Y))$. We further define $[n] = \{1,\ldots,n\}$ for $n \in \mathbb{N}$.

### 2.1 SETTINGS OF THE LATENT VARIABLE MODEL

This work focuses on encoder–decoder models for unsupervised learning, specifically those with discrete latent spaces, including models such as the VQ-VAE (Van Den Oord et al., 2017) and its stochastic extensions (Williams et al., 2020; Takida et al., 2022; Sønderby et al., 2017; Roy et al., 2018). Let $\mathcal{X} \subset \mathbb{R}^d$ be the data space and we assume an unknown data generating distribution $\mathcal{D}$. We express the latent space $\mathcal{Z} \subset \mathbb{R}^{d_z}$, with both $\mathcal{X}$ and $\mathcal{Z}$ equipped with the Euclidean metric $\|\cdot\|$. In the discrete latent space, there are $K$ distinct points, represented as $\mathbf{e} = \{e_j\}_{j=1}^K \in \mathcal{Z}^K$, which are collectively referred to as a codebook learned from the training data, as explained below.

Encoder–decoder models consist of two components: the encoder network $f_\phi : \mathcal{X} \to \mathcal{Z}$ and the decoder network $g_\theta : \mathcal{Z} \to \mathcal{X}$, parameterized by $\phi \in \Phi \subset \mathbb{R}^{d_e}$ and $\theta \in \Theta \subset \mathbb{R}^{d_d}$, respectively. For a given data point $x$, the encoder network transforms it into $f_\phi(x)$ and selects the corresponding discrete representation $e_j$ from the codebook $\mathbf{e}$. The posterior categorical distribution over the index is given as $q(J = j|\mathbf{e}, \phi, x)$ for $j = 1, \ldots, K$. We will introduce examples of this distribution later. Using the selected latent representation $e_J$, the decoder network reconstructs the data as $g_\theta(e_J)$. To generate new data, the index $J$ is drawn from a prior distribution, such as a uniform distribution, and the decoder network returns $g_\theta(e_J)$.

Given the training dataset $S = (S_1, \ldots, S_n) \in \mathcal{X}^n$, where each data point $S_m \in \mathcal{X}$ is independent and identically distributed (i.i.d.) from $\mathcal{D}$, we jointly learn the parameters of the encoder, decoder, and codebook. We denote the set of parameters as $W := \{\mathbf{e}, \phi, \theta\} \in \mathcal{W} := \mathcal{Z}^K \times \Phi \times \Theta$. We assume that these parameters are learned using a randomized algorithm and the learning process is represented by the conditional distribution $\mathbf{e}, \phi, \theta \sim q(\mathbf{e}, \phi, \theta|S)$. The learning algorithm typically minimizes the reconstruction loss. For a given data point $x$ and the corresponding latent variable $e_j$, the quality of the reconstructed data is measured by the loss function $l(x, g_\theta(e_j))$, where $l : \mathcal{X} \times \mathcal{X} \to \mathbb{R}^+$. Then, the reconstruction loss for the input $x$ and parameter $w$ is defined as $l_0 : \mathcal{W} \times \mathcal{X} \to \mathbb{R}, l_0(w, x) := \mathbb{E}_{q(J|\mathbf{e}, \phi, x)} l(x, g_\theta(e_J))$. In this work, we focus on the squared distance for the loss function $l$, so we aim to minimize $l_0(w, x) := \mathbb{E}_{q(J|\mathbf{e}, \phi, x)} \|x - g_\theta(e_J)\|^2$ over the training dataset $x \in S$.

Finally, we provide examples of the posterior distribution $q(J|\mathbf{e}, \phi, x)$, fo the original VQ-VAE (Van Den Oord et al., 2017), the following deterministic mechanism was used:

$$q(J = j|\mathbf{e}, \phi, x) = \begin{cases} 1 & \text{for } j = \arg\min_{k \in [K]} \|f_\phi(x) - e_k\|, \\ 0 & \text{otherwise,} \end{cases} \tag{1}$$

using the distance between the outputs of the encoder and codebook. Recently, stochastic selection methods have become popular. For instance, Williams et al. (2020) proposed the distribution

$$q(J = j|\mathbf{e}, \phi, x) \propto \exp\left(-\beta\|f_\phi(x) - e_j\|^2\right), \tag{2}$$

where the softmax function is used, and $\beta \in \mathbb{R}^+$ is a temperature parameter that controls the level of stochasticity. Beyond this, using stochastic encoders has become common in several other works, including those by Sønderby et al. (2017), Roy et al. (2018), and Takida et al. (2022).

## 2.2 INFORMATION-THEORETIC GENERALIZATION ERROR ANALYSIS

We now briefly outline the IT analysis using the supersample that we utilize in our study (Steinke & Zakynthinou, 2020; Harutyunyan et al., 2021; Hellström & Durisi, 2022). Note that the existing IT analysis is used for supervised learning and the notation in this section is slightly different from our main results in Section 3. Let $\mathcal{X}$ be the domain of data and let us suppose $\mathcal{D}$ represents an *unknown* data distribution. We consider a randomized algorithm $\mathcal{A} : \mathcal{X}^n \to \mathcal{W}$, where $w \in \mathcal{W} \subset \mathbb{R}^{d_w}$ is a parameter. Given the training dataset $S = (S_1, \ldots, S_n)$, the learning algorithm can be characterized by $q(W|S)$. We evaluate the quality of the learning algorithm using the loss function $l : \mathcal{W} \times \mathcal{X} \to [0, 1]$, where $l(\mathcal{A}(s), x)$ is the loss for fixed $S = s$ and $X = x$. Then the training loss is defined as $\hat{L}_S := \frac{1}{n} \sum_{m=1}^n l(\mathcal{A}(S), S_m)$ and the population loss is $L := \mathbb{E}_X l(\mathcal{A}(S), X)$. In the supersample setting, we define $\tilde{X} \in \mathcal{X}^{n \times 2}$ as an $n \times 2$ matrix, where each entry is drawn i.i.d. from $\mathcal{D}$. We refer to this matrix as a *supersample*. Each column of $\tilde{X}$ has the indexes $\{0, 1\}$ associated with $U = (U_1, \ldots, U_n) \sim \text{Uniform}(\{0, 1\}^n)$ independent of $\tilde{X}$. We denote the $m$-th row as $\tilde{X}_m$ with entries $(\tilde{X}_{m,0}, \tilde{X}_{m,1})$. We consider $\tilde{X}_U := (\tilde{X}_{m,U_m})_{m=1}^n$ as the training dataset and $\tilde{X}_{\bar{U}} := (\tilde{X}_{m,\bar{U}_m})_{m=1}^n$ as the test dataset, where $\bar{U}_m = 1 - U_m$. Using the supersamples, the training loss is expressed as $\hat{L}_{\tilde{X}} := \frac{1}{n} \sum_{m=1}^n l(\mathcal{A}(\tilde{X}_U), \tilde{X}_{m,U_m})$ and the test loss is $L_{\tilde{X}} := \frac{1}{n} \sum_{m=1}^n l(\mathcal{A}(\tilde{X}_U), \tilde{X}_{m,\bar{U}_m})$. Furthermore, $l(\mathcal{A}(\tilde{X}_U), \tilde{X})$ denotes the $n \times 2$ loss matrix obtained by applying $l(\mathcal{A}(\tilde{X}_U), \cdot)$ elementwise to $\tilde{X}$. The described setting collectively called the **supersample setting** lead to the following generalization error bound:

**Theorem 1** (Hellström & Durisi (2022)). *Under the supersample setting, we have*

$$|\mathbb{E}_{S,X}(L - \hat{L}_S)| = |\mathbb{E}_{\tilde{X},U}(L_{\tilde{X}} - \hat{L}_{\tilde{X}})| \leq \sqrt{\frac{2}{n} I(l(\mathcal{A}(\tilde{X}_U), \tilde{X}); U|\tilde{X})}.$$

## 3 GENERALIZATION OF THE RECONSTRUCTION LOSS

In this section, we aim to analyze the generalization capability of encoder–decoder models using IT analysis. Following the notation in Section 2.1, given the training dataset $S = (S_1, \ldots, S_n)$, we define the generalization error of the reconstruction loss as

$$\text{gen}(n, \mathcal{D}) := \left| \mathbb{E}_{S,X} \mathbb{E}_{q(\mathbf{e}, \phi, \theta|S)} \left( \mathbb{E}_{q(J|\mathbf{e}, \phi, X)} l(X, g_\theta(e_J)) - \frac{1}{n} \sum_{m=1}^n \mathbb{E}_{q(J_m|\mathbf{e}, \phi, S_m)} l(S_m, g_\theta(e_{J_m})) \right) \right|.$$

To proceed with the analysis, we assume the following condition regarding the data space:

**Assumption 1.** *There exists a positive constant $\Delta$ such that $\sup_{x, x' \in \mathcal{X}} \|x - x'\| < \Delta^{1/2}$.*

This assumption implies that for any $x$, $e_j$, and $\theta$, the loss function $l(x, g_\theta(e_j))$ is bounded by $\Delta$.

We now restate the settings from Section 2.1 under the supersample setting. Given a supersample $\tilde{X} := (\tilde{X}_0, \tilde{X}_1) \in \mathcal{X}^{n \times 2}$, we define $\tilde{X}_U := (\tilde{X}_{m,U_m})_{m=1}^n$ as the training dataset and $\tilde{X}_{\bar{U}} := (\tilde{X}_{m,\bar{U}_m})_{m=1}^n$ as the test dataset. Then, treating $l_0(w, x) := \mathbb{E}_{q(J|\mathbf{e}, \phi, x)} \|x - g_\theta(e_J)\|^2$ as $l$ in Section. 2.2 and rewriting the generalization error using supersample, we can directly apply the generalization bound in Theorem 1. We refer to this generalization bound as the **naive IT-bound** (See Appendix B for the formal statement.). As discussed in Section 1.1, the naive IT-bound does not clearly capture the role of the learned representation $e_J$ in generalization because the CMI term is entangled with both the learning of $W = \{\mathbf{e}, \phi, \theta\}$ and the posterior distribution $q(J|\mathbf{e}, \phi, x)$. See Appendix B for a detailed discussion about this point. In this section, we aim to extend the naive IT analysis to the bound that explicitly captures the role of representation.

### 3.1 THE GENERALIZATION ERROR UNDER THE EXISTING SUPERSAMPLE SETTING

We introduce the notations of the joint distributions used in our theory. Given the supersample $\tilde{X}$, we define $q(\bar{\mathbf{J}}|\mathbf{e}, \phi, \tilde{X}_{\bar{U}}) = \prod_{m=1}^n q(\bar{J}_m|\mathbf{e}, \phi, \tilde{X}_{m,\bar{U}_m})$, $q(\mathbf{J}|\mathbf{e}, \phi, \tilde{X}_U) = \prod_{m=1}^n q(J_m|\mathbf{e}, \phi, \tilde{X}_{m,U_m})$, and $q(\tilde{\mathbf{J}}|\mathbf{e}, \phi, \tilde{X}) = q(\bar{\mathbf{J}}, \mathbf{J}|\mathbf{e}, \phi, \tilde{X}_{\bar{U}}, \tilde{X}_U) = q(\bar{\mathbf{J}}|\mathbf{e}, \phi, \tilde{X}_{\bar{U}}) q(\mathbf{J}|\mathbf{e}, \phi, \tilde{X}_U)$.

The following is our first main result, the proof is shown in Appendix C.

**Theorem 2.** *Under Assumption 1 and the supersample setting, we have*

$$\text{gen}(n, \mathcal{D}) \leq 2\Delta \sqrt{\frac{I(\tilde{\mathbf{J}}; U|\mathbf{e}, \phi, \tilde{X}) + \mathbb{E}_{\tilde{X}, U}\mathbb{E}_{q(\mathbf{e}, \phi, \theta|\tilde{X}_U)}\text{KL}(\mathbf{Q}\|\mathbf{P})}{n}} + \frac{\Delta}{\sqrt{n}}, \quad (3)$$

*where the CMI is defined as*

$$I(\tilde{\mathbf{J}}; U|\mathbf{e}, \phi, \tilde{X}) = \mathbb{E}_{\tilde{X}, U}\mathbb{E}_{q(\mathbf{e}, \phi|\tilde{X}_U)}\text{KL}(q(\tilde{\mathbf{J}}|\mathbf{e}, \phi, \tilde{X})\|\mathbb{E}_{U'}q(\bar{\mathbf{J}}, \mathbf{J}|\mathbf{e}, \phi, \tilde{X}_{\bar{U}'}, \tilde{X}_{U'})). \quad (4)$$

*The distributions of KL divergence are defined as*

$$\mathbf{Q} \coloneqq q(\mathbf{e}, \phi, \theta|\tilde{X}_U) \prod_{m=1}^{n} q(J_m|\mathbf{e}, \phi, \tilde{X}_{m, U_m}), \quad \mathbf{P} \coloneqq q(\mathbf{e}, \phi, \theta|S) \prod_{m=1}^{n} q(J_m|\mathbf{e}, \phi),$$

*and $q(J_m|\mathbf{e}, \phi)$ is any prior distribution that does not depend on the training data.*

The bound does not depend on the decoder's information; This means that we can use a complex decoder network to reduce training reconstruction loss, and it does not worsen the generalization gap. The CMI and KL terms are affected solely by the posterior distribution of the latent variables, conditioned on the learned $\phi$ and $\mathbf{e}$.

**Role of the representation in our bound:** Denoting $\tilde{X}_U = S = (S_1, \ldots, S_n)$, the KL divergence term can be rewritten as

$$\frac{\mathbb{E}_{q(\mathbf{e}, \phi, \theta|S)}\text{KL}(\mathbf{Q}|\mathbf{P})}{n} = \frac{1}{n} \sum_{m=1}^{n} \mathbb{E}_{q(\mathbf{e}, \phi|S)}\text{KL}(q(J_m|\mathbf{e}, \phi, S_m)\|q(J_m|\mathbf{e}, \phi)).$$

This is referred to as **the empirical KL divergence** by Mbacke et al. (2023), which is often used as the regularization in the variational inference. For the CMI term, we have the following relation,

$$I(\tilde{\mathbf{J}}; U|\mathbf{e}, \phi, \tilde{X}) \leq \sum_{m=1}^{n} I(e_J; \tilde{X}_{m, \bar{U}_m}|\mathbf{e}, \phi) + \mathbb{E}_S \mathbb{E}_{q(\mathbf{e}, \phi|S)} \sum_{m=1}^{n} \text{KL}(q(J_m|\mathbf{e}, \phi, S_m)\|q(J_m|\mathbf{e}, \phi)). \quad (5)$$

See Appendix D.1 for its proof. Since $\tilde{X}_{\bar{U}}$ are i.i.d., all $\{I(e_J; \tilde{X}_{m, \bar{U}_m}|\mathbf{e}, \phi)\}_{m=1}^{n}$ are equivalent, thus, we express the first term of Eq. (5) as $\sum_m I(e_J; \tilde{X}_{m, \bar{U}_m}|\mathbf{e}, \phi) = nI(e_J; X|\mathbf{e}, \phi)$. This CMI is commonly used in the IB hypothesis. As pointed out by Sefidgaran et al. (2023), the empirical KL term can be regarded as the "empirical mutual information" $\hat{I}(J; X|\mathbf{e}, \phi)$ by choosing the marginal distribution under $q(J_m|\mathbf{e}, \phi, S_m)$ as the prior distribution. This leads to the generalization bound

$$\text{gen}(n, \mathcal{D}) \leq 2\Delta \sqrt{I(e_J; X|\mathbf{e}, \phi) + 2\mathbb{E}_S \hat{I}(J; X|\mathbf{e}, \phi)} + \frac{\Delta}{\sqrt{n}}. \quad (6)$$

This bound clearly highlights the role of the information encoded in the latent representations in the context of generalization. However, as discussed by Sefidgaran et al. (2023), the bound of Eq. (6) does not vanish as $n \to \infty$. Therefore, the following discussion suggests that utilizing the symmetry of the prior distribution (concerning the supersample) is important to address such issues.

**Dependency on sample size:** Next, we study the dependencies of the CMI and KL terms on $n$ in Theorem 2. The CMI term is similar to the fCMI from the existing IT analysis (Harutyunyan et al., 2021), but here, the conditioning on all other parameters distinguishes it from typical fCMI bounds, see Appendix. D.3 for the detailed discussion. Since the latent space is discrete, we have $I(\tilde{\mathbf{J}}; U|\mathbf{e}, \phi, \tilde{X}) \leq 2n\log K$, ensuring that the bound is always finite, although it may be vacuous. When using the deterministic decoder $f_\phi : \mathcal{X} \to [K]$, we can directly use the existing result (Theorem 8 in Hellström & Durisi (2022)); if $f_\phi$ belongs to a class of functions with a finite Natarajan dimension, $I(\tilde{\mathbf{J}}; U|\mathbf{e}, \phi, \tilde{X}) = \mathcal{O}(\log n)$ (see Appendix D.2 for details). Thus, by regularizing the encoder model's capacity, the first term inside the square root in Theorem 2 scales as $\mathcal{O}(\log n/n)$. Comparing this with Eq. (5), where $I(e_J; X|\phi)$ does not vanish as $n \to \infty$, this highlights the importance of using symmetry in the prior distribution for supersamples to achieve meaningful bounds, as discussed by Sefidgaran et al. (2023). For a stochastic encoder, such as that in Eq. (2), regularizing the encoder's capacity similarly bounds the CMI (see Appendix F and Theorem 4 below).

Regarding the empirical KL term, it is much larger than the CMI term (see Section 5 for the numerical validation), and it does not necessarily vanish as $n \to \infty$, as pointed out by Geiger & Koch (2019) and Sefidgaran et al. (2023). As discussed in Appendix C.3, this arises from the limited flexibility of the supersample setting, which leads to the introduction of the novel supersample setting in Section 3.2.

Finally, we point out that the following data-dependent prior can be used for Theorem 2:

$$\mathbf{P} \coloneqq q(\mathbf{e}, \phi, \theta | S) \prod_{m=1}^{n} \sum_{m'=1}^{n} \frac{1}{N} q(J_m | \mathbf{e}, \phi, S_{m'}).$$

This is an empirical approximation of the marginal distribution, a type of prior that appears in the Vamp prior VAE (Tomczak & Welling, 2018). See Appendix C.3 for the detailed proof.

## 3.2 GENERALIZATION UNDER THE PERMUTATION SYMMETRY SETTINGS

As discussed in Section 3.1, the existing supersample setting leads to an empirical KL term that does not necessarily vanish as $n \to \infty$. As discussed by Sefidgaran et al. (2023), the existing supersample setting utilizes the specific symmetry of the test and training datasets (they referred to it as type-1 symmetry) and demonstrated that such symmetry is insufficient to analyze latent variable models. We extend their results by introducing a new symmetry, which eliminates the empirical KL term.

To establish this new symmetry, let us denote a random permutation of $[2n]$ as $\mathbf{T} = \{T_1, \ldots, T_{2n}\}$, where each permutation appears with uniform probability, $P(\mathbf{T}) = 1/(2n)!$. Given a supersample $\tilde{X} = (\tilde{X}_1, \ldots, \tilde{X}_{2n}) \in \mathcal{X}^{2n}$, a set of $2n$ random variables drawn i.i.d. from $\mathcal{D}$, we reorder the samples using $\mathbf{T}$ expressed as $\tilde{X}_{\mathbf{T}} = (\tilde{X}_{T_1}, \ldots, \tilde{X}_{T_{2n}})$. The first $n$ samples $(\tilde{X}_{T_1}, \ldots, \tilde{X}_{T_n})$ are used for the test dataset and the remaining $n$ samples $(\tilde{X}_{T_{n+1}}, \ldots, \tilde{X}_{T_{2n}})$ are used for the training dataset. We further express $\mathbf{T} = \{\mathbf{T}_0, \mathbf{T}_1\}$, and $\tilde{X}_{\mathbf{T}_0} = (\tilde{X}_{T_1}, \ldots, \tilde{X}_{T_n})$ and $\tilde{X}_{\mathbf{T}_1} = (\tilde{X}_{T_{n+1}}, \ldots, \tilde{X}_{T_{2n}})$ represent the test and training datasets, respectively. Unlike the existing supersample setting discussed in Section 2.2, where $U_m$ are independent, the components of $\mathbf{T}$ are dependent.

We express the joint distibution as follows: $q(\bar{\mathbf{J}} | \mathbf{e}, \phi, \tilde{X}_{\mathbf{T}_0}) = \prod_{m=1}^{n} q(\bar{J}_m | \mathbf{e}, \phi, \tilde{X}_{T_m})$, $q(\mathbf{J} | \mathbf{e}, \phi, \tilde{X}_{\mathbf{T}_1}) = \prod_{m=1}^{n} q(J_m | \mathbf{e}, \phi, \tilde{X}_{T_{n+m}})$, and $q(\tilde{\mathbf{J}} | \mathbf{e}, \phi, \tilde{X}) = q(\bar{\mathbf{J}}, \mathbf{J} | \mathbf{e}, \phi, \tilde{X}_{\mathbf{T}_0}, \tilde{X}_{\mathbf{T}_1}) = q(\bar{\mathbf{J}} | \mathbf{e}, \phi, \tilde{X}_{\mathbf{T}_0}) q(\mathbf{J} | \mathbf{e}, \phi, \tilde{X}_{\mathbf{T}_1})$. We refer to these settings as **the permutation symmetric (supersample) setting**. The following is our main result, and the proof is shown in Appendix E:

**Theorem 3.** *Under Assumptions 1 and the permutation symmetric setting, we have*

$$\mathrm{gen}(n, \mathcal{D}) \leq 4\Delta \mathbb{E}_X \sqrt{\frac{I(\tilde{\mathbf{J}}; \mathbf{T} | \mathbf{e}, \phi, \tilde{X})}{n}} + \frac{2\Delta}{\sqrt{n}},$$

*where the CMI is defined as*

$$I(\tilde{\mathbf{J}}; \mathbf{T} | \mathbf{e}, \phi, \tilde{X}) = \mathop{\mathbb{E}}_{\tilde{X}, \mathbf{T}} \mathop{\mathbb{E}}_{q(\mathbf{e}, \phi | \tilde{X}_{\mathbf{T}_1})} \mathrm{KL}(q(\tilde{\mathbf{J}} | \mathbf{e}, \phi, \tilde{X}) \| \mathop{\mathbb{E}}_{P(\mathbf{T}')} q(\bar{\mathbf{J}}, \mathbf{J} | \mathbf{e}, \phi, \tilde{X}_{\mathbf{T}'_0}, \tilde{X}_{\mathbf{T}'_1})). \tag{7}$$

Compared to Theorem 2, the empirical KL term is eliminated, and a new CMI term, Eq. (7), emerges, which leverages the symmetry of index $\mathbf{T}$ in the prior distribution. So this theorem puts its basis on the careful choice of prior distribution. We will show that this CMI term will vanish as $n \to \infty$; thus, Theorem 3 successfully characterizes the generalization. As discussed in Section 3.1, when using the sufficiently regularized deterministic decoder $f_\phi : \mathcal{X} \to [K]$, this CMI scales as $\mathcal{O}(\log n)$; thus, the bound behaves as $\mathcal{O}(\sqrt{\log n / n})$. See Appendix D.2 for more details.

To analyze the role of the capacity of stochastic encoders such as that in Eq. (2), we extend Theorem 3 by incorporating the concept of *metric entropy*. Assume $q(J | \mathbf{e}, \phi, x) = q(J | \mathbf{e}, f_\phi(x))$. Let $\mathcal{F}$ be the encoder function class equipped with the metric $\| \cdot \|_\infty$. Given $x^n \coloneqq (x_1, \ldots, x_n) \in \mathcal{X}^n$, define the pseudo-metric $d_n$ on $\mathcal{F}$ as $d_n(f, g) \coloneqq \max_{i \in [n]} \|f(x_i) - g(x_i)\|_\infty$ for $f, g \in \mathcal{F}$. The $\delta$-covering number of $\mathcal{F}$ with respect to $d_n$ is denoted as $\mathcal{N}(\delta, \mathcal{F}, x^n)$, and we define $\mathcal{N}(\delta, \mathcal{F}, n) \coloneqq \sup_{x^n \in \mathcal{X}^n} \mathcal{N}(\delta, \mathcal{F}, x^n)$.

**Theorem 4.** *Assume that there exists a positive constant $\Delta_z$ such that $\sup_{z, z' \in \mathcal{Z}} \|z - z'\| < \Delta_z$. Then, when using Eq. (2) and under the same setting as Theorem 3, for any $\delta \in (0, 1]$, we have*

$$\mathrm{gen}(n, \mathcal{D}) \leq \Delta \sqrt{8\beta n \delta \Delta_z} + 4\Delta \sqrt{\frac{\log \mathcal{N}(\delta, \mathcal{F}, 2n)}{n}} + \frac{2\Delta}{\sqrt{n}}.$$

In the proof, we first approximate $f_\phi$ using the $\delta$-cover of $\mathcal{F}$, leading to an approximation error in the first term. Then, the CMI (Eq. (7)) of the $\delta$-cover is bounded by the metric entropy. See Appendix F for the complete proof, including a more general stochastic encoder beyond Eq. (2). When $\mathcal{F}$ is sufficiently regularized (such as with the Natarajan dimension with a margin, see Appendix F for details), the metric entropy scales as $\mathcal{O}(\log(n/\delta))$, and by setting $\delta = \mathcal{O}(1/n^2)$, we achieve $\mathrm{gen}(n, \mathcal{D}) = \mathcal{O}(\sqrt{\log n/n})$. This result demonstrates that regularizing the encoder's capacity leads to better generalization, where as the decoder's capacity does not affect the generalization gap.

## 3.3 DISCUSSION ABOUT THE STOCHASTICITY OF THE ENCODER

Here, we discuss the role of the stochasticity of the encoder in generalization. Proofs of this section are given in Appendix G. Theorems 2 and 3 suggest that, in addition to the encoder's capacity, its stochasticity is critical in enhancing generalization. For instance, in the setting of Theorem 2, let $\hat{Z} = f_\phi(x)$ and $q(J|\mathbf{e}, \phi, x) = q(J|\mathbf{e}, f_\phi(x))$. This formulates the Markov chain $U - \hat{Z} - e_J$ conditioned on $\tilde{X}$. From the data processing inequality (Cover & Thomas, 2012), the CMI between $U$ and $e_J$ is smaller than that between $U$ and $\hat{Z}$. A similar argument holds in Theorem 3, implying that composing the conditional distribution reduces the CMI, thereby improving generalization.

An example of this composition can be found in the stochastic quantized VAE (SQ-VAE) proposed by Takida et al. (2022). In this model, the input data $x$ is compressed as $\hat{Z} = f_\phi(x)$, and then, using a conditional distribution $p(Z_q|\hat{Z})$ (e.g., a Gaussian distribution), a noisy version of $\hat{Z}$ is obtained. Then, the latent variable $e_J$ is obtained using $q(J = j|Z_q, \mathbf{e}) \propto \exp\left(-\beta\|Z_q - e_j\|^2\right)$, which forms the Markov chain $U - \hat{Z} - Z_q - e_J$. From the data processing inequality, the CMI between $U$ and $\hat{Z}$ or $U$ and $Z_q$ is larger than that between $U$ and $e_J$, thus improving generalization.

Next, we examine the stability of the latent variables. The CMI in Eq. (4) is bounded as

$$I(\tilde{\mathbf{J}}; U|\mathbf{e}, \phi, \tilde{X}) \leq \mathop{\mathbb{E}}_{\tilde{X}, U, U'} \mathop{\mathbb{E}}_{q(\mathbf{e}, \phi|\tilde{X}_U)} \mathrm{KL}(q(\bar{\mathbf{J}}, \mathbf{J}|\mathbf{e}, \phi, \tilde{X}_{\bar{U}}, \tilde{X}_U)|q(\bar{\mathbf{J}}, \mathbf{J}|\mathbf{e}, \phi, \tilde{X}_{\bar{U}'}, \tilde{X}_{U'})). \quad (8)$$

The upper bound implies the stability of KL divergence under different data points, suggesting that improving the stability of the posterior distribution of latent variables enhances generalization. Generalizing this, we define the following type of stability; for fixed $\phi$ and $\mathbf{e}$, assume that for all $\mathbf{x}, \mathbf{x}' \in \mathcal{X}$ and for any $j \in [K]$, $q(J = j|\mathbf{e}, \phi, \mathbf{x}) \leq e^\epsilon q(J = j|\mathbf{e}, \phi, \mathbf{x}')$ holds, where $\epsilon \in \mathbb{R}$ may depend on $\phi$ and $\mathbf{e}$. This leads to the $\epsilon$-KL stability, and from Theorem 2, we have

$$\mathrm{gen}(n, \mathcal{D}) \leq 2\Delta\sqrt{3\mathbb{E}_S\mathbb{E}_{q(\mathbf{e}, \phi|S)}\epsilon} + \frac{\Delta}{\sqrt{n}}. \quad (9)$$

This result illustrates the importance of the stability of the latent variables. The introduced stability is conceptually similar to differential privacy (DP) (Dwork et al., 2006), but with a key difference; whereas DP is defined for all datasets, the stability here applies to individual data points. This distinction arises because the posterior distribution is conditionally independent of each data point.

Note that the posterior distribution in Eq. (2) corresponds to the exponential mechanism in the privacy context (McSherry & Talwar, 2007). Conditioned on $\phi$ and $\mathbf{e}$, assume that for any $x \in \mathcal{X}$ and any $j \in [K]$, there exists $\Delta_{\phi, \mathbf{e}} \in \mathbb{R}^+$ such that $\|f_\phi(x) - e_j\|^2 \leq \Delta_{\phi, \mathbf{e}}$. Then Eq. (2) satisfies the stability condition of Eq. (9) with $\epsilon = 2\beta\Delta_{\phi, \mathbf{e}}$ and we have $\mathrm{gen}(n, \mathcal{D}) \leq 2\Delta\sqrt{3\beta\mathbb{E}_S\mathbb{E}_{q(\mathbf{e}, \phi|S)}\Delta_{\phi, \mathbf{e}}} + \frac{\Delta}{\sqrt{n}}$. This bound provides a natural interpretation of the temperature parameter $\beta$, which controls the level of stochasticity. As $\beta \to \infty$, Eq. (2) becomes deterministic, causing the bound to become vacuous. On the other hand, if $\beta \to 0$, Eq. (2) approaches a uniform distribution that ignores the input data, thereby improving generalization with an increased reconstruction loss on the training data.

## 3.4 COMPARISON WITH EXISTING BOUNDS

Here, we compare our bounds with those in existing work. Theorem 2 resembles the results of Mbacke et al. (2023) since both bounds include the empirical KL term in the upper bounds, and the posterior distribution corresponds to the variational posterior distribution. The key difference is that Mbacke et al. (2023) assumed fixed encoder and decoder parameters, whereas our analysis incorporates the learning process under the assumption of a finite latent space and a squared reconstruction loss.

Another distinction is that their generalization bound does not become $0$ as $n \to \infty$ due to two reasons. One is the presence of the empirical KL term, which we address in Theorem 3 using permutation symmetry. Our technique can be regarded as developing the appropriate prior distribution in PAC-Bayes bound. The second reason is the presence of the average distance $\frac{1}{n}\sum_{m=1}^{n} \mathbb{E}_X \|X - S_m\|$ in the existing bound, which is inherent to the data distribution and may not vanish as $n \to \infty$. Our use of the squared loss in the analysis mitigates this problematic term, as detailed in Appendix C.

Our proof techniques are based on Sefidgaran et al. (2023). However, we could not directly apply their methods, as the reconstruction loss reuses input data, unlike in classification settings. We resolve this by combining the data regeneration technique used in the proof of Mbacke et al. (2023). Additionally, we introduced a new permutation symmetric setting, leading to a bound that controls mutual information in Theorem 3. Our setting is closely related to the type-2 symmetry proposed in Sefidgaran et al. (2023), which involves random permutations selecting $n$ indices from $2n$ with a uniform distribution $1/\binom{2n}{n}$, whereas our setting requires the consideration of the order of the permutation index to evaluate the exponential moment (see Appendix E). Finally, we theoretically studied the behavior of the CMI (Theorem 4) focusing on the complexity of the encoder, whereas Sefidgaran et al. (2023) provided the bounds based on the CMI without such discussion.

The existing analyses based on the IB hypothesis (Vera et al., 2018; Hafez-Kolahi et al., 2020; Kawaguchi et al., 2023; Vera et al., 2023) assumed that both the latent variables and data are discrete, and their obtained bounds explicitly depend on the latent space size or show exponential dependence on the MI. In contrast, we assume that only latent variables are discrete and the resulting bound does not explicitly depend on the number of discrete states nor exhibit exponential dependence on MI.

## 4    DATA GENERATION GUARANTEE FOR THE ENCODER−DECODER MODEL

The primary interest of latent variable models often lies in their data generation capability rather than their generalization under the reconstruction loss. Specifically, the aim is to generate realistic data by sampling from the latent variable distribution and transforming it via the decoder. We expect that the generated data distribution is close to the true data distribution.

Let $p$ represent a distribution on $\mathcal{Z}$, and let us assume that for any $\theta \in \Theta$, the decoder $g_\theta(\cdot): \mathcal{Z} \to \mathcal{X}$ is measurable. The pushforward of the distribution $p$ by the decoder, denoted as $g_\theta \# p$, defines a distribution on $\mathcal{X}$ as $g_\theta \# p(A) = p(g_\theta^{-1}(A))$ for any measurable set $A \subseteq \mathcal{X}$. When generating data, we first draw an index using the prior distribution $p(J|\mathbf{e}, \phi)$, which is typically independent of the training dataset. This corresponds to selecting a latent variable $e_J$ from $\{e_1, \ldots, e_K\}$, and we denote the associated prior distribution over $\mathcal{Z}$ as $p(e|\mathbf{e}, \phi)$. The resulting generated data distribution is then represented as $\hat{\mu} := g_\theta \# p(e|\mathbf{e}, \phi)$. Next, given the posterior distribution $q(J_m|\mathbf{e}, \phi, S_m)$ conditioned on the $m$-th training data point $S_m$, we express the corresponding posterior distribution over $\mathcal{Z}$ as $q(e_{(m)}|\mathbf{e}, \phi, S_m)$, where we simply express $e_{J_m}$ as $e_{(m)}$. Here, we evaluate the 2-Wasserstein distance (see Appendix A for the definition) between the data distribution $\mathcal{D}$ and the generated data distribution $\hat{\mu}$, denoted as $W_2(\mathcal{D}, \hat{\mu})$. The following is our main result:

**Theorem 5.** *Let $S = (S_1, \ldots, S_n) \in \mathcal{X}^n$ be a training dataset, where $S_m \in \mathcal{X}$ are drawn i.i.d. from $\mathcal{D}$. Under Assumption 1 and for any prior $q(e|\mathbf{e}, \phi)$ that does not depend on $S$, we have*

$$\mathbb{E}_S \mathbb{E}_{q(\mathbf{e},\phi,\theta|S)} W_2^2(\mathcal{D}, \hat{\mu}) \leq \mathbb{E}_S \mathbb{E}_{q(\mathbf{e},\phi,\theta|S)} \frac{2}{n} \sum_{m=1}^{n} \mathbb{E}_{q(e_{(m)}|\mathbf{e},\phi,S_m)} \|S_m - g_\theta(e_{(m)})\|^2$$

$$+ 2\Delta \sqrt{\frac{2}{n} \sum_{m=1}^{n} \mathbb{E}_S \mathbb{E}_{q(\mathbf{e},\phi|S)} \mathrm{KL}(q(e_{(m)}|\mathbf{e},\phi,S_m) \| q(e_{(m)}|\mathbf{e},\phi))} + \frac{2\Delta}{\sqrt{n}}.$$

This theorem shows that the 2-Wasserstein distance is upper-bounded by the reconstruction loss on the training dataset and an empirical KL term. The result is similar to the bound obtained by Mbacke et al. (2023), where the fixed parameters are assumed, that is, learning is not considered as discussed in Section 1.1. In contrast, our bound incorporates the learning process of parameters. If the marginal distribution of $q(e|\mathbf{e}, \phi, x)$ were used as the prior distribution, the empirical KL term would become the empirical MI as discussed in Section 3.1. Furthermore, if a prior distribution with the symmetry introduced in Section 3.2 were used, the empirical KL term would become the CMI appearing in

Theorem 3. However, such priors are impractical in real-world scenarios, where uniform distributions are typically used to sample latent variables.

Theorems 2, 3, and 5 offer important insights into the roles of the encoder and decoder. To improve the generalization and data generation capabilities, it is desirable to use a complex decoder, as it can lower the training reconstruction loss without increasing the KL or CMI terms in the upper bound, regardless of the sample size. However, using the complex encoder increases the KL and CMI, requiring careful adjustment according to the sample size. This characteristic is specific to latent variable models, highlighting the critical role of the latent variables as the regularization.

## 5 EXPERIMENTS

In this section, we present experimental results on MNIST (LeCun et al., 1989) to analyze the behavior of the CMI term ($I(\check{\mathbf{J}}; U|\mathbf{e}, \phi, \tilde{X})/n$) and the KL term ($\mathbb{E}_{\tilde{X}, U}\mathbb{E}_{q(\mathbf{e}, \phi, \theta|\tilde{X}_U)}\mathrm{KL}(\mathbf{Q}\|\mathbf{P})/n$) in our generalization error bound and to confirm the validity of our provided in Theorem 2 and to confirm the validity of the discussions based on it. These values were empirically evaluated using 15 models trained under five supersample settings prepared based on three different combinations of training and test datasets. We also measured the *generalization gap*, which is the absolute value of the difference in the empirical reconstruction loss calculated using the training and test data as the estimated value of the generalization error. In our experiments, we adopted the SQ-VAE model proposed by Takida et al. (2022) with the ConvResNets architecture. The SQ-VAE utilizes a posterior distribution similar to Eq. (1), based on the Gumbel-Softmax relaxation Jang et al. (2017); Maddison et al. (2017), enabling efficient optimization of discrete latent variables and resulting in the excellent generalization performance. The details of our experimental settings are provided in Appendix I.

We first conducted experiments to elucidate the behavior of the CMI and KL divergence terms in our bound as $n$ increases. Here, we adopted the following sample-size settings: $n = \{250, 1000, 2000, 4000\}$. The results are presented in Figure 1. From these results, it is evident that the CMI term (center plot) decreases as $n$ increases. Moreover, we observe a correlation between the generalization gap (left plot) and the CMI term. In addition, the CMI term values are quite close to the generalization gap. In contrast, the values of the KL divergence term (right plot), which employ a uniform distribution as the prior, take extremely large values, do not correlate well with the generalization gap, and do not necessarily decrease as $n$ increases. This observation aligns with our discussion in Section 3.1 regarding the importance of evaluating the generalization error through the construction of an appropriate prior using supersamples.

Next, to analyze the behavior of the CMI and KL term in response to changes in model complexity, we fixed the number of training samples at $n = 4000$ and evaluated models trained with $K = \{16, 32, 64, 128\}$. The results are presented in Figure 2. This result indicates that our bound increases only by $\log K$. In contrast, existing studies (Hafez-Kolahi et al., 2021) show that generalization bounds depend exponentially on MI or entropy, which possibly shows $\mathcal{O}(K)$. These differences highlight the superiority of our bound.

## 6 CONCLUSION AND LIMITATIONS

We conducted the first comprehensive analysis of the generalization and data generation capabilities of encoder–decoder models in unsupervised learning based on the IT analysis. Our work highlights the role of encoder capacity and the posterior distribution of latent variables through a novel permutation symmetric supersample setting. However, our analysis has two limitations. First, it assumes a discrete latent space, limiting its applicability to models such as VAEs with continuous latent variables. Second, it relies on the squared loss for reconstruction. Addressing these limitations in future work is crucial for developing a more accurate understanding of encoder–decoder models. Furthermore, from our observations in Section 5, our bound expressed by the CMI term without the KL term in Theorem 3 may be more reasonable compared to that in Theorem 2. However, it is challenging to estimate $I(\tilde{\mathbf{J}}; \mathbf{T}|\mathbf{e}, \phi, \tilde{X})$ since $\mathbf{T}$ is a high-dimensional random variable ($2n$ dimensions), which prevent us verifying its validity through numerical experiments. Exploring estimation methods for such a CMI term with high-dimensional random variables, or deriving a bound based solely on the CMI through alternative approaches, constitutes an important direction for future research.

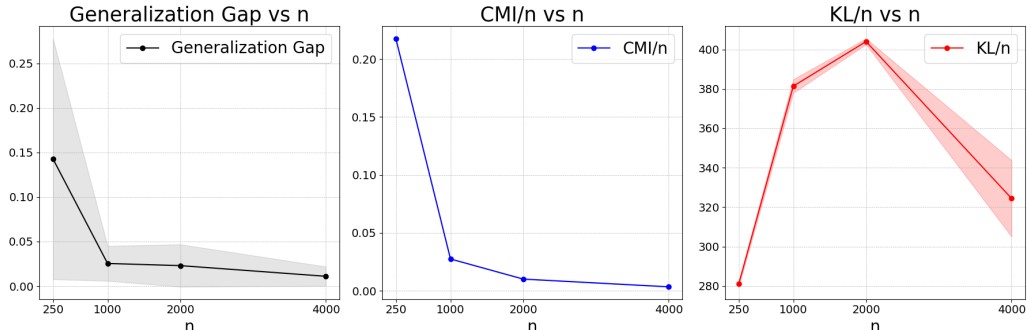

Figure 1: The behavior of the generalization gap (left), the CMI term ($I(\tilde{\mathbf{J}}; U|\mathbf{e}, \phi, \tilde{X})/n$) (center), and the KL term ($\mathbb{E}_{\tilde{X}, U}\mathbb{E}_{q(\mathbf{e}, \phi, \theta|\tilde{X}_U)}\mathrm{KL}(\mathbf{Q}\|\mathbf{P})/n$) (right) when the number of training samples $n$ is increasing ($K = 128, d_z = 64$).

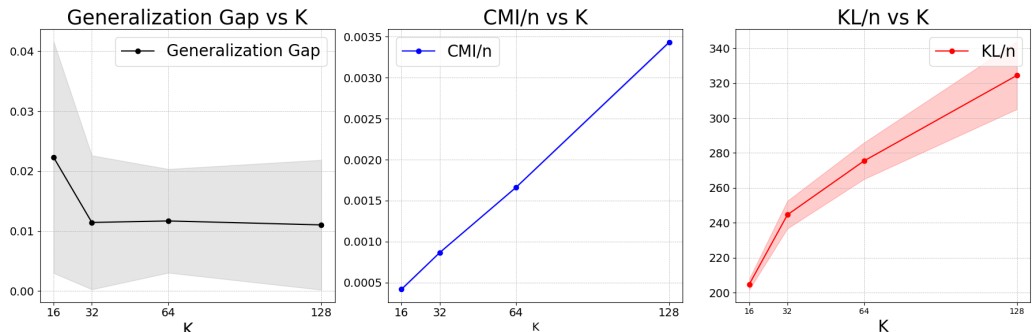

Figure 2: The behavior of the generalization gap (left), the CMI term ($I(\tilde{\mathbf{J}}; U|\mathbf{e}, \phi, \tilde{X})/n$) (center), and the KL term ($\mathbb{E}_{\tilde{X}, U}\mathbb{E}_{q(\mathbf{e}, \phi, \theta|\tilde{X}_U)}\mathrm{KL}(\mathbf{Q}\|\mathbf{P})/n$) (right) when the size of a codebook $K$ is increasing ($n = 4000, d_z = 64$).

## ETHICS STATEMENT

As this paper primarily focuses on theoretical analysis, there are no notable ethical risks, and our work does not require IRB approval. Although our experiments are included to validate the theory numerically, there is no concern regarding the responsible release of data or models with high misuse potential. We used widely available benchmark datasets, which are publicly available.

## REPRODUCIBILITY STATEMENT

The primary focus of this study is the theoretical analysis of unsupervised learning. All theoretical claims are clearly stated with their respective assumptions, and detailed derivations of all theorems and equations are provided in the Appendix. While the study includes numerical experiments, their purpose is to validate the theory empirically. For reproducibility, we used publicly available benchmark datasets. Moreover, the models and evaluation methods employed in this study are based on publicly available code from existing research (Takida et al., 2022). We have summarized the details of our experimental settings in Appendix I, including the URLs for the source code we downloaded, hyperparameter settings, and information about the computational resources used. In addition, we will make our experimental source code publicly available on GitHub upon publication of this paper. Therefore, the reproducibility of our numerical experiments is adequately ensured.

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

## A    AUXILIARY DEFINITIONS AND LEMMAS

Here we define the Wasserstein distance. Given a metric $d(\cdot, \cdot)$ and probability distributions $p$ and $q$ on $\mathcal{X}$, let $\Pi(p, q)$ denote the set of all couplings of $p$ and $q$. The 2-Wasserstein distance is defined as:

$$W_2(p, q) = \sqrt{\inf_{\rho \in \Pi} \int_{\mathcal{X} \times \mathcal{X}} d(x, x')^2 d\rho(x, x').}$$

In this work, we use the Euclidean metric $|\cdot|$ as $d(\cdot, \cdot)$.

We also rely on the following type of exponential moment inequality, which is often used in the proof of McDiarmid's inequality. A function $f : \mathcal{X}^n \to \mathbb{R}$ has the bounded differences property if for some nonnegative constants $c_1, \ldots, c_n$, the following holds for all $i$:

$$\sup_{x_1, \ldots, x_n, x_i' \in \mathcal{X}} |f(x_1, \ldots, x_n) - f(x_1, \ldots, x_{i-1}, x_i', x_{i+1}, \ldots, x_n)| \le c_i, \quad 1 \le i \le n.$$

Assuming $X_1, \ldots, X_n$ are independent random variables taking values in $\mathcal{X}$, we have the following lemma:

**Lemma 1** (Used in the proof of McDiarmid's inequality). *Given a function $f$ with the bounded differences property, for any $t \in \mathbb{R}$, we have:*

$$\mathbb{E}\left[e^{t(f(X_1, \ldots, X_n) - \mathbb{E}[f(X_1, \ldots, X_n)])}\right] \le e^{\frac{t^2}{8} \sum_{i=1}^n c_i^2}.$$

## B    DISCUSSION ABOUT THE NAIVE IT BOUND

As discussed in Sec 3, by applying the existing IT analysis bound in Theorem 1, we can derive a naive IT bound for the reconstruction loss as follows:

**Theorem 6.** *Under Assumption 1 and the supersample setting, we have*

$$\text{gen}(n, \mathcal{D}) \leq \Delta \sqrt{\frac{2}{n} I(l_0(W, \tilde{X}); U | \tilde{X})}.$$

*where $l_0(w, x) := \mathbb{E}_{q(J|\mathbf{e}, \phi, x)} \|x - g_\theta(e_J)\|^2$ and $W = \{\mathbf{e}, \phi, \theta\} \sim q(\mathbf{e}, \phi, \theta | \tilde{X}_U)$.*

*Proof.* Note that the generalization error can be expressed as the supersample

$$\text{gen}(n, \mathcal{D})$$

$$= \left| \mathbb{E}_{S,X} \mathbb{E}_{q(\mathbf{e}, \phi, \theta | S)} \left( \mathbb{E}_{q(J|\mathbf{e}, \phi, X)} l(X, g_\theta(e_J)) - \frac{1}{n} \sum_{m=1}^{n} \mathbb{E}_{q(J_m|\mathbf{e}, \phi, S_m)} l(S_m, g_\theta(e_{J_m})) \right) \right|$$

$$= \left| \underset{\tilde{X}, U}{\mathbb{E}} \underset{q(\mathbf{e}, \phi, \theta | X_U)}{\mathbb{E}} \left( \frac{1}{n} \sum_{m=1}^{n} \mathbb{E}_{q(\bar{J}_m|\mathbf{e}, \phi, X_{m, \bar{U}_m})} l(X_{m, \bar{U}_m}, g_\theta(e_{J_m})) \right. \right.$$

$$\left. \left. - \frac{1}{n} \sum_{m=1}^{n} \mathbb{E}_{q(J_m|\mathbf{e}, \phi, X_{m, U_m})} l((X_{m, U_m}, g_\theta(e_{J_m}))) \right) \right|$$

Given that the loss is bounded by $[0, \Delta]$, it follows a $\Delta$-subGaussian property. Thus, using Theorem 1, we obtain the result. $\square$

It is important to note that this upper bound is characterized by the CMI $I(l_0(W, \tilde{X}); U | \tilde{X})$. This CMI depends on the decoder and encoder information, distinguishing it from the results presented in our main Theorems 2 and 3, which do not require the decoder's information.

To clarify this distinction, let us introduce the necessary notation. Following the notation in Section 3.1, we define the regenerated data as:

$$\tilde{Y} := (g_\theta(e_{\bar{J}1}), \dots, g_\theta(e_{\bar{J}n}), g_\theta(e_{J1}), \dots, g_\theta(e_{Jn})) = g_\theta(e_{\tilde{J}}),$$

which represents the elementwise application of the decoder $g_\theta(e_{(\cdot)})$ to the selected index $\tilde{J}$ on $\tilde{X}$ (Recall that $q(\tilde{J}|\mathbf{e}, \phi, \tilde{X}) = q(\bar{J}, J|\mathbf{e}, \phi, \tilde{X}_{\bar{U}}, \tilde{X}_U) = q(\bar{J}|\mathbf{e}, \phi, \tilde{X}_{\bar{U}}) q(J|\mathbf{e}, \phi, \tilde{X}_U)$ .).

Under these notations, we have the following relations:

$$I(l_0(W, \tilde{X}); U | \tilde{X}) \leq I(\tilde{Y}; U | \tilde{X}) \leq I(\theta; U | \tilde{X}) + I(e_{\tilde{J}}; U | \tilde{X}, \theta)$$

where the first inequality is obtained by the data processing inequality (DPI) and the second inequality is obtained by the chain rule of CMI and the DPI. This result demonstrates that the decoder information cannot be eliminated from the naive IT bound, which clarifies the fundamental difference compared to our result (Theorems 2 and 3). Moreover, since the decoder and encoder are learned simultaneously using the same training data, they are not independent. This makes it unclear how the latent variables and the encoder's capacity affect generalization, as it is difficult to eliminate the decoder's dependency on them.

## C  PROOF OF THEOREM 2

We express $q(\tilde{J}|\mathbf{e}, \phi, \tilde{X}) = q(\bar{J}, J|\mathbf{e}, \phi, \tilde{X}_{\bar{U}}, \tilde{X}_U) = q(\bar{J}|\mathbf{e}, \phi, \tilde{X}_{\bar{U}}) q(J|\mathbf{e}, \phi, \tilde{X}_U)$. Hereinafter, we simplify the notation by expressing $\tilde{X}$ as $X$. For simplification in the proof, we omit the absolute operation for the generalization gap. The reverse bound can be proven in a similar manner. We first

express the generalization error of the reconstruction loss using the supersample as follows

$$\sum_{k=1}^{K} \frac{1}{n} \sum_{m=1}^{n} \mathbb{E}_{q(\bar{J}_m|\mathbf{e},\phi,X_{m,\bar{U}_m})q(\mathbf{e},\phi,\theta|X_U)} l(X_{m,\bar{U}_m}, g_\theta(e_k)) \mathbb{1}_{k=\bar{J}_m}$$

$$-\sum_{k=1}^{K} \frac{1}{n} \sum_{m=1}^{n} \mathbb{E}_{q(J_m|\mathbf{e},\phi,X_{m,U_m})q(\mathbf{e},\phi,\theta|X_U)} l((X_{m,U_m}, g_\theta(e_k)) \mathbb{1}_{k=J_m}$$

$$=\sum_{k=1}^{K} \frac{1}{n} \sum_{m=1}^{n} \mathbb{E}_{q(\bar{J}_m|\mathbf{e},\phi,X_{m,\bar{U}_m})q(\mathbf{e},\phi,\theta|X_U)} \|X_{m,\bar{U}_m} - g_\theta(e_k)\|^2 \mathbb{1}_{k=\bar{J}_m}$$

$$-\sum_{k=1}^{K} \frac{1}{n} \sum_{m=1}^{n} \mathbb{E}_{q(J_m|\mathbf{e},\phi,X_{m,U_m})q(\mathbf{e},\phi,\theta|X_U)} \|X_{m,U_m} - g_\theta(e_k)\|^2 \mathbb{1}_{k=J_m}, \quad (10)$$

where the first term corresponds to the test loss and the second term corresponds to the training loss.

Recall the learning algorithm and posterior distribution:

$$\mathbf{e}, \phi, \theta \sim q(\mathbf{e}, \phi, \theta|X_U),$$
$$j_k \sim q(\mathbf{J}|\mathbf{e}, \phi, x_k).$$

Here $\mathbf{e} = \{e_1, \ldots, e_K\}$ is the codebook, and $j$ and $\mathbf{J} = \{J_1, \ldots, j_n\}$ represents the index of the codebook that the test and training data are represented.

Conditioned on $X$ and $U$, we then decompose Eq. (10) as follows

$$\sum_{k=1}^{K} \frac{1}{n} \sum_{m=1}^{n} \mathbb{E}_{q(\mathbf{e},\phi,\theta|X_U)} l(X_{m,\bar{U}_m}, g_\theta(e_k)) \mathbb{E}_{q(\bar{J}_m|\mathbf{e},\phi,X_{m,\bar{U}_m})} \mathbb{1}_{k=\bar{J}_m}$$

$$-\sum_{k=1}^{K} \frac{1}{n} \sum_{m=1}^{n} \mathbb{E}_{q(\mathbf{e},\phi,\theta|X_U)} l(X_{m,\bar{U}_m}, g_\theta(e_k)) \mathbb{E}_{q(J_m|\mathbf{e},\phi,X_{m,U_m})} \mathbb{1}_{k=J_m}$$

$$+\sum_{k=1}^{K} \frac{1}{n} \sum_{m=1}^{n} \mathbb{E}_{q(\mathbf{e},\phi,\theta|X_U)} l(X_{m,\bar{U}_m}, g_\theta(e_k)) \mathbb{E}_{q(J_m|\mathbf{e},\phi,X_{m,U_m})} \mathbb{1}_{k=J_m}$$

$$-\sum_{k=1}^{K} \frac{1}{n} \sum_{m=1}^{n} \mathbb{E}_{q(\mathbf{e},\phi,\theta|X_U)} l(X_{m,U_m}, g_\theta(e_k)) \mathbb{E}_{q(J_m|\mathbf{e},\phi,X_{m,U_m})} \mathbb{1}_{k=J_m}. \quad (11)$$

We will separately upper bound these terms.

## C.1 Bounding first and second terms

The decomposition of the generalization error, as shown in Eq. (11), allows us to bound the first and second terms as follows.

We apply Donsker-Varadhan's inequality between the following two distributions:

$$\mathbf{Q} \coloneqq P(U)q(\mathbf{e}, \phi, \theta|X_U)q(\bar{\mathbf{J}}, \mathbf{J}|\mathbf{e}, \phi, X_{\bar{U}}, X_U)$$
$$\mathbf{P}_S \coloneqq P(U)q(\mathbf{e}, \phi, \theta|X_U) \mathop{\mathbb{E}}_{P(U')} q(\bar{\mathbf{J}}, \mathbf{J}|\mathbf{e}, \phi, X_{\bar{U}'}, X_{U'}). \quad (12)$$

Then, for any $\lambda \in \mathbb{R}^+$, we have

$$\sum_{k=1}^{K} \frac{1}{n} \sum_{m=1}^{n} \mathbb{E}_{q(\mathbf{e},\phi,\theta|X_U)} l(X_{m,\bar{U}_m}, g_\theta(e_k)) \left( \mathbb{E}_{q(\bar{J}_m|\mathbf{e},\phi,X_{m,\bar{U}_m})} \mathbb{1}_{k=\bar{J}_m} - \mathbb{E}_{q(J_m|\mathbf{e},\phi,X_{m,U_m})} \mathbb{1}_{k=J_m} \right)$$

$$\leq \frac{1}{\lambda} \mathrm{KL}(\mathbf{Q}|\mathbf{P}_S) + \frac{1}{\lambda} \log \mathbb{E}_{\mathbf{P}_S} \exp \left( \frac{\lambda}{n} \sum_{k=1}^{K} \sum_{m=1}^{n} l(X_{m,\bar{U}_m}, g_\theta(e_k)) \left( \mathbb{1}_{k=\bar{J}_m} - \mathbb{1}_{k=J_m} \right) \right).$$

To simplify the notation, we express $\bar{\mathbf{J}} = \mathbf{J}_0$, $\bar{J}_m = J_{m,0}$, $\mathbf{J} = \mathbf{J}_1$, and $J_m = J_{m,1}$. Let $U''$ be a random variable taking $0, 1$ with a uniform distribution. Since $\mathbf{P}_S$ is symmetric with respect to the permutation of $\mathbf{J}_0$ and $\mathbf{J}_1$, we can bound the exponential moment as:

$$\log \mathbb{E}_{P(U)q(\mathbf{e},\phi,\theta|X_U)\underset{P(U')}{\mathbb{E}} q(\mathbf{J}_0,\mathbf{J}_1|\mathbf{e},\phi,X_{\bar{U}'},X_{U'})} \exp\left(\frac{\lambda}{n}\sum_{k=1}^{K}\sum_{m=1}^{n} l(X_{m,\bar{U}_m}, g_\theta(e_k))\left(\mathbb{1}_{k=J_{m,0}} - \mathbb{1}_{k=J_{m,1}}\right)\right)$$

$$= \log \mathbb{E}_{P(U)q(\mathbf{e},\phi,\theta|X_U)P(U'')^n\underset{P(U')}{\mathbb{E}} q(\mathbf{J}_0,\mathbf{J}_1|\mathbf{e},\phi,X_{\bar{U}'},X_{U'})P(U'')^N}$$

$$\exp\left(\frac{\lambda}{n}\sum_{k=1}^{K}\sum_{m=1}^{n} l(X_{m,\bar{U}_m}, g_\theta(e_k))\left(\mathbb{1}_{k=J_{m,\bar{U}''}} - \mathbb{1}_{k=J_{m,U''}}\right)\right)$$

$$= \log \mathbb{E}_{P(U)q(\mathbf{e},\phi,\theta|X_U)\underset{P(U')}{\mathbb{E}} q(\mathbf{J}_0,\mathbf{J}_1|\mathbf{e},\phi,X_{\bar{U}'},X_{U'})\mathbb{E}_{P(U'')^n}}$$

$$\exp\left(\frac{\lambda}{n}\sum_{k=1}^{K}\sum_{m=1}^{n} l(X_{m,\bar{U}_m}, g_\theta(e_k))\left(\mathbb{1}_{k=J_{m,\bar{U}''}} - \mathbb{1}_{k=J_{m,U''}}\right)\right).$$

In the final line, we apply McDiarmid's inequality since $U''^n$ are $n$ i.i.d. random variables. To use McDiarmid's inequality in Lemma 1, we use the stability caused by replacing one of the elements of $n$ i.i.d. random variables. To estimate the coefficients of stability in Lemma 1, let $U''^n = (U''_1, \ldots, U''_N)$, then

$$\sup_{\{U''_m\}_{m=1}^n, U'''_{m'}} \left| \frac{\lambda}{n}\sum_{k=1}^{K}\sum_{m=1}^{n} l(X_{m,\bar{U}_m}, g_\theta(e_k))\left(\mathbb{1}_{k=J_{m,\bar{U}''_m}} - \mathbb{1}_{k=J_{m,U''_m}}\right) \right. \tag{13}$$

$$- \frac{\lambda}{n}\sum_{k=1}^{K}\sum_{m\neq m'}^{n} l(X_{m,\bar{U}_m}, g_\theta(e_k))\left(\mathbb{1}_{k=J_{m,\bar{U}''_m}} - \mathbb{1}_{k=J_{m,U''_m}}\right)$$

$$\left. - \frac{\lambda}{n}\sum_{k=1}^{K} l(X_{m',\bar{U}'_m}, g_\theta(e_k))\left(\mathbb{1}_{k=J_{m',\bar{U}'''_{m'}}} - \mathbb{1}_{k=J_{m',U'''_{m'}}}\right) \right|$$

$$= \sup_{\{U''_m\}_{m=1}^n, U'''_{m'}} \left| \frac{\lambda}{n}\sum_{k=1}^{K} l(X_{m',\bar{U}'_m}, g_\theta(e_k))\left(\mathbb{1}_{k=J_{m',\bar{U}''_m}} - \mathbb{1}_{k=J_{m',U''_m}}\right) \right.$$

$$\left. - \frac{\lambda}{n}\sum_{k=1}^{K} l(X_{m',\bar{U}'_m}, g_\theta(e_k))\left(\mathbb{1}_{k=J_{m',\bar{U}'''_m}} - \mathbb{1}_{k=J_{m',U'''_m}}\right) \right| \leq \frac{2\lambda\Delta}{n}$$

Here, the maximum change caused by replacing one element of $U''$ is $2\lambda\Delta/n$, thus, its log of the exponential moment is bounded by $(2\lambda\Delta/n)^2/8 \times n = \lambda^2\Delta^2/2n$. Thus from Lemma 1, we have

$$\log \mathbb{E}_{P(U)q(\mathbf{e},\phi,\theta|X_U)\underset{P(U')}{\mathbb{E}} q(\mathbf{J}_0,\mathbf{J}_1|\mathbf{e},\phi,X_{\bar{U}'},X_{U'})} \exp\left(\frac{\lambda}{n}\sum_{k=1}^{K}\sum_{m=1}^{n} l(X_{m,\bar{U}_m}, g_\theta(e_k))\left(\mathbb{1}_{k=J_{m,0}} - \mathbb{1}_{k=J_{m,1}}\right)\right)$$

$$\leq \frac{\lambda^2\Delta^2}{2n}.$$

Finally, by noting that

$$\mathbb{E}_X \mathrm{KL}(\mathbf{Q}|\mathbf{P}_S)$$

$$= \mathbb{E}_X \underset{P(U)q(\mathbf{e},\phi|X_U)}{\mathbb{E}} \mathrm{KL}(q(\bar{\mathbf{J}},\mathbf{J}|\mathbf{e},\phi,X_{\bar{U}},X_U)|\mathbb{E}_{U'}q(\bar{\mathbf{J}},\mathbf{J}|\mathbf{e},\phi,X_{\bar{U}'},X_{U'})) = I(\bar{\mathbf{J}},\mathbf{J};U|\mathbf{e},\phi,X),$$

the first and second terms in Eq. (11) are upper bounded by

$$\frac{1}{\lambda}I(\bar{\mathbf{J}},\mathbf{J};U|\mathbf{e},\phi,X) + \frac{\lambda\Delta^2}{2n}. \tag{14}$$

## C.2 BOUNDING THIRD AND FOURTH TERMS

Next, we upper bound the third and fourth terms in Eq. (11);

$$
\sum_{k=1}^{K} \frac{1}{n} \sum_{m=1}^{n} \mathbb{E}_{q(\mathbf{e},\phi,\theta|X_U)} l(X_{m,\bar{U}_m}, g_\theta(e_k)) \mathbb{E}_{q(J_m|\mathbf{e},\phi,X_{m,U_m})} \mathbb{1}_{k=J_m}
$$

$$
- \sum_{k=1}^{K} \frac{1}{n} \sum_{m=1}^{n} \mathbb{E}_{q(\mathbf{e},\phi,\theta|X_U)} l(X_{m,U_m}, g_\theta(e_k)) \mathbb{E}_{q(J_m|\mathbf{e},\phi,X_{m,U_m})} \mathbb{1}_{k=J_m}. \tag{15}
$$

We simplify the notation by expressing $\mathbb{E}_{q(J_m|\mathbf{e},\phi,X_{m,U_m})} \mathbb{1}_{k=J_m}$ as $P_{k,m}$ and use the square loss:

$$
\mathbb{E}_{X,U} \sum_{k=1}^{K} \frac{1}{n} \sum_{m=1}^{n} \mathbb{E}_{q(\mathbf{e},\phi,\theta|X_U)} l(X_{m,\bar{U}_m}, g_\theta(e_k)) P_{k,m} - \sum_{k=1}^{K} \frac{1}{n} \sum_{m=1}^{n} \mathbb{E}_{q(\mathbf{e},\phi,\theta|X_U)} l(X_{m,U_m}, g_\theta(e_k)) P_{k,m}
$$

$$
= \mathbb{E}_{X,U} \sum_{k=1}^{K} \frac{1}{n} \sum_{m=1}^{n} \mathbb{E}_{q(\mathbf{e},\phi,\theta|X_U)} \left( \|X_{m,\bar{U}_m}\|^2 - \|X_{m,U_m}\|^2 \right) P_{k,m}
$$

$$
+ \mathbb{E}_{X,U} \sum_{k=1}^{K} \frac{2}{n} \sum_{m=1}^{n} \mathbb{E}_{q(\mathbf{e},\phi,\theta|X_U)} \left( X_{m,\bar{U}_m} - X_{m,U_m} \right) \cdot g_\theta(e_k) P_{k,m}
$$

$$
= \mathbb{E}_{X,U} \frac{1}{n} \sum_{m=1}^{n} \left( \|X_{m,\bar{U}_m}\|^2 - \|X_{m,U_m}\|^2 \right) \mathbb{E}_{q(\mathbf{e},\phi,\theta|X_U)} \sum_{k=1}^{K} P_{k,m}
$$

$$
+ \mathbb{E}_S \frac{2}{n} \sum_{m=1}^{n} \left( \mathbb{E}_X X - X_m \right) \cdot \mathbb{E}_{q(\mathbf{e},\phi,\theta|S)} \sum_{k=1}^{K} g_\theta(e_k) P_{k,m}
$$

$$
= \mathbb{E}_S \frac{2}{n} \sum_{m=1}^{n} \left( \mathbb{E}_X X - X_m \right) \cdot \mathbb{E}_{q(\mathbf{e},\phi,\theta|S)} \sum_{k=1}^{K} g_\theta(e_k) P_{k,m}, \tag{16}
$$

where we express $S = (X_{1,U_1}, \ldots, X_{n,U_n}) = (S_1, \ldots, S_n)$ as the training samples. In the last inequality, we used $\sum_{k=1}^{K} P_{k,m} = 1$ and $\mathbb{E}_{X,U} \frac{1}{n} \sum_{m=1}^{n} \left( \|X_{m,\bar{U}_m}\|^2 - \|X_{m,U_m}\|^2 \right) = 0$ since $X$ and $U$ are i.i.d.

To evaluate the final line, we use the Donsker-Valadhan inequality between

$$
\mathbf{Q} := q(\mathbf{e}, \phi, \theta|S) \prod_{m=1}^{n} q(J_m|\mathbf{e}, \phi, S_m),
$$

$$
\mathbf{P}_S := q(\mathbf{e}, \phi, \theta|S) \prod_{m=1}^{n} q(J_m|\mathbf{e}, \phi),
$$

where $q(J_m|\mathbf{e}, \phi)$ is the prior distribution, which never depends on the training data. Then we have

$$
\mathbb{E}_S \frac{2}{n} \sum_{m=1}^{n} \left( \mathbb{E}_X X - X_m \right) \cdot \mathbb{E}_{q(\mathbf{e},\phi,\theta|S)} \sum_{k=1}^{K} g_\theta(e_k) P_{k,m}
$$

$$
\leq \mathbb{E}_S \frac{1}{\lambda} \mathrm{KL}(\mathbf{Q}|\mathbf{P}_S) + \mathbb{E}_S \frac{1}{\lambda} \log \mathbb{E}_{\mathbf{P}_S} \exp \left( \frac{2\lambda}{n} \sum_{m=1}^{n} \left( \mathbb{E}_X X - X_m \right) \cdot \mathbb{E}_{q(\mathbf{e},\phi,\theta|S)} \sum_{k=1}^{K} g_\theta(e_k) \mathbb{1}_{k=J_m} \right)
$$

$$
\leq \mathbb{E}_S \frac{1}{\lambda} \mathrm{KL}(\mathbf{Q}|\mathbf{P}_S)
$$

$$
+ \mathbb{E}_S \frac{1}{\lambda} \log \mathbb{E}_{\mathbf{P}_S} \exp \left( \frac{2\lambda}{n} \sum_{m=1}^{n} \left( \mathbb{E}_X X - X_m \right) \cdot \sum_{k=1}^{K} g_\theta(e_k) (\mathbb{1}_{k=J_m} - P''_{k,m}) \right)
$$

$$
+ \mathbb{E}_S \mathbb{E}_{\mathbf{P}_S} \frac{2}{n} \sum_{m=1}^{n} \left( \mathbb{E}_X X - X_m \right) \cdot \sum_{k=1}^{K} g_\theta(e_k) P''_{k,m}, \tag{17}
$$

where $P''_{k,m} = \mathbb{E}_{q(J_m|\phi,\mathbf{e})}\mathbb{1}_{k=J_m}$. Clearly, this does not depend on the index $m$, so we express $P''_{k,m} = P''_k$. Then the last term becomes

$$\mathbb{E}_S\mathbb{E}_{\mathbf{P}_S}\frac{1}{n}\sum_{m=1}^{n}(\mathbb{E}_X X - X_m)\cdot\sum_{k=1}^{K}g_\theta(e_k)P''_k \leq \mathbb{E}_S\mathbb{E}_{\mathbf{P}_S}\left\|\mathbb{E}_X X - \frac{1}{n}\sum_{m=1}^{n}X_m\right\|\,\|\sum_{k=1}^{K}g_\theta(e_k)P''_k\|$$

$$\leq \mathbb{E}_S\left\|\mathbb{E}_X X - \frac{1}{n}\sum_{m=1}^{n}X_m\right\|\sqrt{\Delta}$$

$$\leq \sqrt{\Delta\mathrm{Var}\left(\frac{1}{n}\sum_{m=1}^{n}X_m\right)}$$

$$\leq \sqrt{\Delta\frac{\mathrm{Var}\,(X)}{n}}$$

$$\leq \sqrt{\frac{\Delta}{4n}}\sqrt{\Delta} = \frac{\Delta}{2\sqrt{n}}, \tag{18}$$

where we used the fact that the variance of random variables with bounded in $(a, b]$ is upper bounded by $(b-a)^2/4n$ (the extension to the $d$-dimensional random variable is straightforward) and thus, $\mathrm{Var}\,(X) \leq \Delta/4$. Then the exponential moment term becomes

$$\mathbb{E}_S\frac{1}{\lambda}\log\mathbb{E}_{\mathbf{P}_S}\exp\left(\frac{2\lambda}{n}\sum_{m=1}^{n}(\mathbb{E}_X X - X_m)\cdot\sum_{k=1}^{K}g_\theta(e_k)(\mathbb{1}_{k=J_m} - P''_{k,m})\right)$$

$$= \mathbb{E}_S\frac{1}{\lambda}\log\mathbb{E}_{\mathbf{P}_S}\exp\left(\frac{2\lambda}{n}\sum_{m=1}^{n}(\mathbb{E}_X X - X_m)\cdot\sum_{k=1}^{K}g_\theta(e_k)(\mathbb{1}_{k=J} - P''_k)\right).$$

Here we use the McDiarmid's inequality for $n$ random variables $\mathbf{J}$. Then we estimate the stability coefficient similarly to Eq. (13), which is upper bounded by $\lambda\Delta/n$. Then from Lemma 1, the exponential moment is bounded by $(2\lambda\Delta/n)^2/8 \times n = \lambda\Delta^2/2n$ Thus, the second term is upper bounded by

$$\frac{1}{\lambda}\mathrm{KL}(\mathbf{Q}|\mathbf{P}_S) + \frac{\lambda\Delta^2}{2n} + \frac{\Delta}{\sqrt{n}}. \tag{19}$$

By optimizing the first and second terms of Eqs. (14) and (19), we have

$$2\Delta\sqrt{\frac{(I(\bar{\mathbf{J}},\mathbf{J};U|\mathbf{e},\phi,X) + \mathbb{E}_S\mathbb{E}_{q(\mathbf{e},\phi,\theta|S)}\mathrm{KL}(\mathbf{Q}|\mathbf{P}_S))}{n}} + \frac{\Delta}{\sqrt{n}},$$

where we used the fact that $X_m$ are i.i.d. Thus, we use McDiarmid's inequality for $n$ random variables of $X_m$ to upper bound the exponential moment. We estimate the stability coefficient similarly to Eq. (13), which is upper bounded by as follows. where

$$\mathbf{Q} := q(\mathbf{e},\phi,\theta|S)\prod_{m=1}^{n}q(J_m|\mathbf{e},\phi,S_m),$$

$$\mathbf{P}_S := q(\mathbf{e},\phi,\theta|S)\prod_{m=1}^{n}q(J_m|\mathbf{e},\phi).$$

## C.3 Discussion about the limitation of the existing supersample setting

The empirical KL divergence in Theorem 2 originates from the third and fourth terms of Eq. (11), as discussed in Appendix C.2. After applying the Donsker-Valadhan lemma in the proof, it is crucial to ensure that the probability $P''_{k,m}$ does not depend on the sample index $m$ to control the exponential moment in Eq. (17). To achieve this, we employ the prior distribution $q(J_m|\mathbf{e},\phi)$, which eliminates

the sample index dependency and leads to $P''_{k,m} = P''_k$. As a result, we can use a distribution of the form:

$$\mathbf{P}_S := q(\mathbf{e}, \phi, \theta | S) \prod_{m=1}^{n} \sum_{m'=1}^{n} \frac{1}{N} q(J_m | \mathbf{e}, \phi, S_{m'}),$$

which provides an empirical approximation of the marginal distribution using available samples. Since this distribution does not explicitly depend on the sample index, we can bound the exponential moment similarly as done in Appendix C.2.

However, using the prior distribution in Eq. (12) to bound the third and fourth terms of Eq. (11) is not feasible. The reason is that applying the Donsker-Valadhan lemma with Eq. (12) to these terms does not yield a bound of order $\mathcal{O}(1/\sqrt{n})$ as achieved in Eq. (18). This is because the dependency on the sample index in Eq. (12) prevents us from leveraging the symmetry between the test and training datasets through the supersample index $U$. Consequently, the prior distribution's symmetry cannot be exploited to simplify the bounds for these terms.

# D PROOF OF LEMMAS AND EQUATIONS

## D.1 PROOF OF EQ. (5)

We define $q(\bar{\mathbf{J}}|\mathbf{e}, \phi) = \prod_{m=1}^{n} q(\bar{J}_m | \mathbf{e}, \phi)$, $q(\mathbf{J}|\mathbf{e}, \phi) = \prod_{m=1}^{n} q(J_m | \mathbf{e}, \phi)$, and $q(\tilde{\mathbf{J}}|\mathbf{e}, \phi, \tilde{X}) = q(\bar{\mathbf{J}}, \mathbf{J}|\mathbf{e}, \phi) = q(\bar{\mathbf{J}}|\mathbf{e}, \phi)q(\mathbf{J}|\mathbf{e}, \phi)$ where each $q(\bar{J}_m | \mathbf{e}, \phi)$ is the marginal distribution of $q(J_m | \mathbf{e}, \phi, X_m)$.

Then by the definition of the CMI, we have

$I(\tilde{\mathbf{J}}; U | \mathbf{e}, \phi, \tilde{X})$

$= \mathbb{E}_{\tilde{X}, U} \mathbb{E}_{q(\mathbf{e}, \phi | \tilde{X}_U)} \mathrm{KL}(q(\tilde{\mathbf{J}} | \mathbf{e}, \phi, \tilde{X}) \| \mathbb{E}_{U'} q(\bar{\mathbf{J}}, \mathbf{J} | \mathbf{e}, \phi, \tilde{X}_{\bar{U}'}, \tilde{X}_{U'}))$

$\leq \mathbb{E}_{\tilde{X}, U} \mathbb{E}_{q(\mathbf{e}, \phi | \tilde{X}_U)} \mathrm{KL}(q(\tilde{\mathbf{J}} | \mathbf{e}, \phi, \tilde{X}) \| q(\bar{\mathbf{J}}, \mathbf{J} | \mathbf{e}, \phi))$

$= \mathbb{E}_{\tilde{X}, U} \mathbb{E}_{q(\mathbf{e}, \phi | \tilde{X}_U)} \mathrm{KL}(q(\bar{\mathbf{J}} | \mathbf{e}, \phi, \tilde{X}_{\bar{U}}) \| q(\bar{\mathbf{J}} | \mathbf{e}, \phi)) + \mathbb{E}_{\tilde{X}, U} \mathbb{E}_{q(\mathbf{e}, \phi | \tilde{X}_U)} \mathrm{KL}(q(\mathbf{J} | \mathbf{e}, \phi, \tilde{X}_U) \| q(\mathbf{J} | \mathbf{e}, \phi))$

$= \mathbb{E}_{\tilde{X}, U} \mathbb{E}_{q(\mathbf{e}, \phi | \tilde{X}_U)} \sum_{m=1}^{n} \mathrm{KL}(q(\bar{J}_m | \mathbf{e}, \phi, \tilde{X}_{m, \bar{U}_m}) \| q(\bar{J}_m | \mathbf{e}, \phi))$

$\quad + \mathbb{E}_{\tilde{X}, U} \mathbb{E}_{q(\mathbf{e}, \phi | \tilde{X}_U)} \sum_{m=1}^{n} \mathrm{KL}(q(J_m | \mathbf{e}, \phi, \tilde{X}_{m, U_m}) \| q(J_m | \mathbf{e}, \phi))$

$= n I(J; X | \mathbf{e}, \phi) + \mathbb{E}_S \mathbb{E}_{q(\mathbf{e}, \phi | S)} \frac{1}{n} \sum_{m=1}^{n} \mathrm{KL}(q(J_m | \mathbf{e}, \phi, S_m) \| q(J_m | \mathbf{e}, \phi))$

$\leq n I(e_J; X | \mathbf{e}, \phi) + \mathbb{E}_S \mathbb{E}_{q(\mathbf{e}, \phi | S)} \frac{1}{n} \sum_{m=1}^{n} \mathrm{KL}(q(J_m | \mathbf{e}, \phi, S_m) \| q(J_m | \mathbf{e}, \phi)).$

## D.2 DISCUSSION ABOUT THE CMI OF THE DETERMINISTIC ENCODER

Here, we consider the case where $f_\phi : \mathcal{X} \to [K]$ represents a deterministic encoder that maps input data to one of the $K$ indices. This scenario can be interpreted as a $K$-class classification problem, allowing us to directly apply the results from Harutyunyan et al. (2021). In their work, they demonstrated that the CMI for multi-class classification problems can be upper-bounded using the Natarajan dimension. The Natarajan dimension is a combinatorial measure that generalizes the VC dimension to a multiclass classification setting. Using this concept, we can derive the following characterization:

When using a deterministic encoder network $f_\phi : \mathcal{X} \to [K]$, belonging to a class with finite Natarajan dimension $d_K$, and assuming $2n > d_K + 1$, we have the following bound:

$$I(\tilde{\mathbf{J}}; U | \mathbf{e}, \phi, \tilde{X}) \leq d_K \log \left( \binom{K}{2} \frac{2en}{d_K} \right). \tag{20}$$

The proof follows exactly as in Theorem 8 of Harutyunyan et al. (2021).

Thus, by regularizing the capacity of the encoder model (via the Natarajan dimension), the CMI term scales as $\mathcal{O}(\log n)$, ensuring controlled generalization behavior. Examples of models that satisfy the finite Natarajan dimension are shown in Jin (2023) and Daniely et al. (2011). Also, see Bendavid et al. (1995), which shows that the VC dimension of the multiclass loss function characterizes the graph dimension, and the graph dimension upper bounds the Natarajan dimension. For the discussion of the stochastic encoder that uses $q(J|\mathbf{e}, \phi, x) = q(J|\mathbf{e}, f_\phi(x))$, see Appendix F.2.

Finally, we remark the CMI of Eq. (7). We show that

$$I(\tilde{\mathbf{J}}; \mathbf{T}|\mathbf{e}, \phi, \tilde{X}) \leq d_K \log\left(\binom{K}{2} \frac{2en}{d_K}\right). \tag{21}$$

By the definition of the CMI, the CMI is expressed as the difference of entropy and conditional entropy. Since $\tilde{J}$ is discrete, the entropy is always larger than 0. Thus, we have

$$I(\tilde{\mathbf{J}}; \mathbf{T}|\mathbf{e}, \phi, \tilde{X}) \leq H[\tilde{\mathbf{J}}|\mathbf{e}, \phi, \tilde{X}].$$

where $H$ is the Shannon entropy. Note that the entropy is bounded by the growth function, i.e., the maximum number of different ways in which a dataset of size $2n$ can be classified in $K$. And such quantity is bounded in the proof of Theorem 8 of Harutyunyan et al. (2021), thus Eq. (21) holds similarly to Eq. (20).

Thus, by regularizing the capacity of the encoder model (via the Natarajan dimension), the CMI term of Eq. (7) scales as $\mathcal{O}(\log n)$.

### D.3 Comparison with the fCMI

Here, we examine the relationship between our CMI and existing forms of fCMI in more detail. As highlighted in the main paper, a key difference is that our CMI is conditioned on all model parameters, whereas existing fCMI approaches marginalize the parameters.

To explore this further, we consider marginalizing over the encoder parameter, $\phi$. In the proof of Theorem 2, we perform this marginalization over $\phi$ in Eq. (10), and obtain

$$\sum_{k=1}^{K} \frac{1}{n} \sum_{m=1}^{n} \mathbb{E}_{q(\bar{J}_m|\mathbf{e}, \phi, X_{m,\bar{U}_m}) q(\mathbf{e}, \phi, \theta|X_U)} l(X_{m,\bar{U}_m}, g_\theta(e_k)) \mathbb{1}_{k=\bar{J}_m}$$

$$- \sum_{k=1}^{K} \frac{1}{n} \sum_{m=1}^{n} \mathbb{E}_{q(J_m|\mathbf{e}, \phi, X_{m,U_m}) q(\mathbf{e}, \phi, \theta|X_U)} l((X_{m,U_m}, g_\theta(e_k)) \mathbb{1}_{k=J_m}$$

$$= \sum_{k=1}^{K} \frac{1}{n} \sum_{m=1}^{n} \mathbb{E}_{q(\bar{J}_m|\theta, \mathbf{e}, X_{m,\bar{U}_m}) q(\mathbf{e}, \theta|X_U)} \|X_{m,\bar{U}_m} - g_\theta(e_k)\|^2 \mathbb{1}_{k=\bar{J}_m}$$

$$- \sum_{k=1}^{K} \frac{1}{n} \sum_{m=1}^{n} \mathbb{E}_{q(J_m|\theta, \mathbf{e}, X_{m,U_m}) q(\mathbf{e}, \theta|X_U)} \|X_{m,U_m} - g_\theta(e_k)\|^2 \mathbb{1}_{k=J_m},$$

and proceed with the proof in the same way. We apply the Donsker-Varadhan inequality between the following distributions, instead of Eq. (12):

$$\mathbf{Q} \coloneqq P(U)P(U')q(\mathbf{e}, \theta|X_U)q(\bar{\mathbf{J}}, \mathbf{J}|, \mathbf{e}, \theta, X_{\bar{U}}, X_U)$$

$$\mathbf{P} \coloneqq P(U)q(\mathbf{e}, \theta|X_U)\mathbb{E}_{P(U')}q(\bar{\mathbf{J}}, \mathbf{J}|\mathbf{e}, \theta, X_{\bar{U}'}, X_{U'}).$$

This incorporates marginalization over $\phi$ in Eq. (12), resulting in the following KL divergence in the upper bound:

$$\mathbb{E}_X \mathrm{KL}(\mathbf{Q}|\mathbf{P}) = \mathbb{E}_X \underset{P(U)q(\mathbf{e},\phi|X_U)}{\mathbb{E}} \mathrm{KL}(q(\bar{\mathbf{J}}, \mathbf{J}|\mathbf{e}, \theta, X_{\bar{U}}, X_U)|\mathbb{E}_{P(U')}q(\bar{\mathbf{J}}, \mathbf{J}|\mathbf{e}, \theta, X_{\bar{U}'}, X_{U'}))$$

$$= I(\bar{\mathbf{J}}, \mathbf{J}; U|\mathbf{e}, \theta, X).$$

Unlike Theorem 2, this CMI explicitly involves the decoder parameter $\theta$. By marginalizing over $\phi$, decoder information is integrated into the upper bound, making Theorem 2 distinct from existing fCMI bounds.

# E  PROOF OF THEOREM 3

We define $\mathbf{T} = \{\mathbf{T}_0, \mathbf{T}_1\}$, where $\tilde{X}_{\mathbf{T}_0} = (\tilde{X}_{T_1}, \ldots, \tilde{X}_{T_n})$ serves as the test dataset and $\tilde{X}_{\mathbf{T}_1} = (\tilde{X}_{T_{n+1}}, \ldots, \tilde{X}_{T_{2n}})$ serves as the training dataset. We further express $\tilde{X}_{\mathbf{T}_0} = (\tilde{X}_{T_1}, \ldots, \tilde{X}_{T_n}) = (\tilde{X}_{\mathbf{T}_{0,1}}, \ldots, \tilde{X}_{\mathbf{T}_{0,n}})$ and $\tilde{X}_{\mathbf{T}_1} = (\tilde{X}_{\mathbf{T}_{1,1}}, \ldots, \tilde{X}_{\mathbf{T}_{1,n}})$. To emphasize the dependence of the dataset on $\mathbf{T}$, we write the posterior distribution as $q(\tilde{\mathbf{J}}|\mathbf{e}, \phi, \tilde{X}_{\mathbf{T}}) = q(\bar{\mathbf{J}}, \mathbf{J}|\mathbf{e}, \phi, \tilde{X}_{\mathbf{T}}) = q(\bar{\mathbf{J}}, \mathbf{J}|\mathbf{e}, \phi, \tilde{X}_{\mathbf{T}_0}, \tilde{X}_{\mathbf{T}_1}) = q(\bar{\mathbf{J}}|\mathbf{e}, \phi, \tilde{X}_{\mathbf{T}_0})q(\mathbf{J}|\mathbf{e}, \phi, \tilde{X}_{\mathbf{T}_1})$.

Hereinafter, we express $\tilde{X}$ as $X$ to simplify the notation. Under the permutation symmetric settings, the generalization error can be expressed as

$$
\underset{S,X}{\mathbb{E}}\, \mathbb{E}_{q(\mathbf{e},\phi,\theta|S)} \left( \mathbb{E}_{q(J|\mathbf{e},\phi,X)}l(X, g_\theta(e_J)) - \frac{1}{n}\sum_{m=1}^n \mathbb{E}_{q(J_m|\mathbf{e},\phi,S_m)}l(S_m, g_\theta(e_{J_m})) \right)
$$

$$
= \mathbb{E}_{X,\mathbf{T}} \sum_{k=1}^K \frac{1}{n}\sum_{m=1}^n \mathbb{E}_{q(\bar{J}_m|\mathbf{e},\phi,X_{\mathbf{T}_{0,m}})q(\mathbf{e},\phi,\theta|X_{\mathbf{T}_1})} l((X_{\mathbf{T}_{0,m}}, g_\theta(e_k))\mathbb{1}_{k=\bar{J}_m}
$$

$$
- \mathbb{E}_{X,\mathbf{T}} \sum_{k=1}^K \frac{1}{n}\sum_{m=1}^n \mathbb{E}_{q(J_m|\mathbf{e},\phi,X_{\mathbf{T}_{1,m}})q(\mathbf{e},\phi,\theta|X_{\mathbf{T}_1})} l(X_{\mathbf{T}_{1,m}}, g_\theta(e_k))\mathbb{1}_{k=J_m}
$$

$$
= \mathbb{E}_{X,\mathbf{T}} \sum_{k=1}^K \frac{1}{n}\sum_{m=1}^n \mathbb{E}_{q(\bar{J}_m|\mathbf{e},\phi,X_{\mathbf{T}_{0,m}})q(\mathbf{e},\phi,\theta|X_{\mathbf{T}_1})} \|X_{\mathbf{T}_{0,m}} - g_\theta(e_k)\|^2 \mathbb{1}_{k=\bar{J}_m}
$$

$$
- \mathbb{E}_{X,\mathbf{T}} \sum_{k=1}^K \frac{1}{n}\sum_{m=1}^n \mathbb{E}_{q(J_m|\mathbf{e},\phi,X_{\mathbf{T}_{1,m}})q(\mathbf{e},\phi,\theta|X_{\mathbf{T}_1})} \|X_{\mathbf{T}_{1,m}} - g_\theta(e_k)\|^2 \mathbb{1}_{k=J_m}. \tag{22}
$$

We then decompose the loss as follows

$$
\mathrm{gen}(n, \mathcal{D}) \tag{23}
$$

$$
= \mathbb{E}_{X,\mathbf{T}} \sum_{k=1}^K \frac{1}{n}\sum_{m=1}^n \mathbb{E}_{q(\bar{J}_m|\mathbf{e},\phi,X_{\mathbf{T}_{0,m}})q(\mathbf{e},\phi,\theta|X_{\mathbf{T}_1})} \|X_{\mathbf{T}_{0,m}} - g_\theta(e_k)\|^2 \mathbb{1}_{k=\bar{J}_m}
$$

$$
- \mathbb{E}_{X,\mathbf{T}} \sum_{k=1}^K \frac{1}{n}\sum_{m=1}^n \mathbb{E}_{q(J_m|\mathbf{e},\phi,X_{\mathbf{T}_{1,m}})q(\mathbf{e},\phi,\theta|X_{\mathbf{T}_1})} \|X_{\mathbf{T}_{0,m}} - g_\theta(e_k)\|^2 \mathbb{1}_{k=J_m}
$$

$$
+ \mathbb{E}_{X,\mathbf{T}} \sum_{k=1}^K \frac{1}{n}\sum_{m=1}^n \mathbb{E}_{q(J_m|\mathbf{e},\phi,X_{\mathbf{T}_{1,m}})q(\mathbf{e},\phi,\theta|X_{\mathbf{T}_1})} \|X_{\mathbf{T}_{0,m}} - g_\theta(e_k)\|^2 \mathbb{1}_{k=J_m}
$$

$$
- \mathbb{E}_{X,\mathbf{T}} \sum_{k=1}^K \frac{1}{n}\sum_{m=1}^n \mathbb{E}_{q(J_m|\mathbf{e},\phi,X_{\mathbf{T}_{1,m}})q(\mathbf{e},\phi,\theta|X_{\mathbf{T}_1})} \|X_{\mathbf{T}_{1,m}} - g_\theta(e_k)\|^2 \mathbb{1}_{k=J_m}.
$$

First, we upper bound the first two terms by applying the Donsker-Varadhan inequality. Consider the joint distribution and the prior distribution, defined as follows:

$$
\mathbf{Q} := P(\mathbf{T})q(\mathbf{e}, \theta, \phi|X_{\mathbf{T}_1})q(\bar{\mathbf{J}}, \mathbf{J}|\mathbf{e}, \phi, X_{\mathbf{T}}),
$$

$$
\mathbf{P} := P(\mathbf{T})q(\mathbf{e}, \theta, \phi|X_{\mathbf{T}_1}) \underset{P(\mathbf{T}')}{\mathbb{E}}\, q(\bar{\mathbf{J}}, \mathbf{J}|\mathbf{e}, \phi, X_{\mathbf{T}'}).
$$

Then we then obtain

$$
\mathbb{E}_{X,\mathbf{T}} \sum_{k=1}^K \frac{1}{n}\sum_{m=1}^n \mathbb{E}_{q(\mathbf{e},\phi,\theta|X_{\mathbf{T}_1})} \|X_{\mathbf{T}_{0,m}} - g_\theta(e_k)\|^2 \left( \mathbb{E}_{q(\bar{J}_m|\mathbf{e},\phi,X_{\mathbf{T}_{1,m}})} \mathbb{1}_{k=\bar{J}_m} - \mathbb{E}_{q(J_m|\mathbf{e},\phi,X_{\mathbf{T}_{0,m}})} \mathbb{1}_{k=J_m} \right)
$$

$$
\leq \mathbb{E}_X \frac{1}{\lambda}\mathrm{KL}(\mathbf{Q}|\mathbf{P}) + \mathbb{E}_X \frac{1}{\lambda} \log \mathbb{E}_{\mathbf{P}} \exp\left( \frac{\lambda}{n}\sum_{k=1}^K\sum_{m=1}^n \|X_{\mathbf{T}_{0,m}} - g_\theta(e_k)\|^2 \left( \mathbb{1}_{k=\bar{J}_m} - \mathbb{1}_{k=J_m} \right) \right).
$$

Note that $\underset{P(\mathbf{T}')}{\mathbb{E}} q(\bar{\mathbf{J}}, \mathbf{J}|\mathbf{e}, \phi, X_{\mathbf{T}'})$ is symmetric with respect to the permutation of $\mathbf{T}$. Thus, we have

$$\log \mathbb{E}_{P(\mathbf{T})q(\mathbf{e},\theta,\phi|X_{\mathbf{T}_1})} \underset{P(\mathbf{T}')}{\mathbb{E}} q(\bar{\mathbf{J}},\mathbf{J}|\mathbf{e},\phi,X_{\mathbf{T}'}) \exp\left(\frac{\lambda}{n}\sum_{k=1}^{K}\sum_{m=1}^{n} l(X_{\mathbf{T}_{0,m}}, g_\theta(e_k))\left(\mathbb{1}_{k=\bar{J}_m} - \mathbb{1}_{k=J_m}\right)\right)$$

$$= \log \mathbb{E}_{P(\mathbf{T})q(\mathbf{e},\theta,\phi|X_{\mathbf{T}_1})} \underset{P(\mathbf{T}')}{\mathbb{E}} q(\bar{\mathbf{J}},\mathbf{J}|\mathbf{e},\phi,X_{\mathbf{T}'})P(\mathbf{T}'')$$

$$\exp\left(\frac{\lambda}{n}\sum_{k=1}^{K}\sum_{m=1}^{n} l(X_{\mathbf{T}_{0,m}}, g_\theta(e_k))\left(\mathbb{1}_{k=J_{\mathbf{T}''_{0,m}}} - \mathbb{1}_{k=J_{\mathbf{T}''_{1,m}}}\right)\right)$$

$$= \log \mathbb{E}_{P(\mathbf{T})q(\mathbf{e},\theta,\phi|X_{\mathbf{T}_1})} \underset{P(\mathbf{T}')}{\mathbb{E}} q(\bar{\mathbf{J}},\mathbf{J}|\mathbf{e},\phi,X_{\mathbf{T}'})$$

$$\mathbb{E}_{P(\mathbf{T}'')}\exp\left(\frac{\lambda}{n}\sum_{k=1}^{K}\sum_{m=1}^{n} l(X_{\mathbf{T}_{0,m}}, g_\theta(e_k))\left(\mathbb{1}_{k=J_{\mathbf{T}''_{0,m}}} - \mathbb{1}_{k=J_{\mathbf{T}''_{1,m}}}\right)\right).$$

To simplify the notation, we define $\mathbf{T}'' = \{\mathbf{T}''_0, \mathbf{T}''_1\} = \{\mathbf{T}''_{0,1}, \ldots, \mathbf{T}''_{0,n}, \mathbf{T}''_{1,1}, \ldots, \mathbf{T}''_{1,n}\}$. Note that $\mathbf{T}''_{j,m}$ for $m = 1, \ldots, n$ and $j = 0, 1$ are not independent of each other due to the permutation that generates them. Therefore, we cannot directly apply standard concentration inequalities, as is possible in the existing supersample setting.

To address this, we use the results from Joag-Dev & Proschan (1983), which concern the negative association of permutation variables. From Theorem 2.11 in Joag-Dev & Proschan (1983), the distribution $P(\mathbf{T})$ satisfies negative association. Additionally, as discussed in Section 3.3 of Joag-Dev & Proschan (1983) and further in Proposition 4 and 5 of Dubhashi & Ranjan (1996), we have that

$$\log \mathbb{E}_{P(\mathbf{T})q(\mathbf{e},\theta,\phi|X_{\mathbf{T}_1})} \underset{P(\mathbf{T}')}{\mathbb{E}} q(\bar{\mathbf{J}},\mathbf{J}|\mathbf{e},\phi,X_{\mathbf{T}'})$$

$$\mathbb{E}_{P(\mathbf{T}'')}\exp\left(\frac{\lambda}{n}\sum_{k=1}^{K}\sum_{m=1}^{n} l(X_{\mathbf{T}_{0,m}}, g_\theta(e_k))\left(\mathbb{1}_{k=J_{\mathbf{T}''_{0,m}}} - \mathbb{1}_{k=J_{\mathbf{T}''_{1,m}}}\right)\right)$$

$$\leq \log \mathbb{E}_{P(\mathbf{T})q(\mathbf{e},\theta,\phi|X_{\mathbf{T}_1})} \underset{P(\mathbf{T}')}{\mathbb{E}} q(\bar{\mathbf{J}},\mathbf{J}|\mathbf{e},\phi,X_{\mathbf{T}'})$$

$$\mathbb{E}_{\prod_{m=1}^{n}\prod_{j=0,1}P(\mathbf{T}''_{j,m})}\exp\left(\frac{\lambda}{n}\sum_{k=1}^{K}\sum_{m=1}^{n} l(X_{\mathbf{T}_{0,m}}, g_\theta(e_k))\left(\mathbb{1}_{k=J_{\mathbf{T}''_{0,m}}} - \mathbb{1}_{k=J_{\mathbf{T}''_{1,m}}}\right)\right),$$

where $P(\mathbf{T}''_{j,m})$ is the marginal distribution, implying that $\mathbf{T}''_{j,m}$ are now $2n$ independent random variables. Intuitively, the results in Joag-Dev & Proschan (1983) indicate that the elements of the permutation index, which follow the permutation distribution, are negatively correlated. As a result, the expectation of the marginal distribution is larger than that of the joint distribution.

Since $\{\mathbf{T}''_{j,m}\}$ are independent, we can apply McDiarmid's inequality, which leads to the results in

$$\log \mathbb{E}_{P(\mathbf{T})q(\mathbf{e},\theta,\phi|X_{\mathbf{T}_1})} \underset{P(\mathbf{T}')}{\mathbb{E}} q(\bar{\mathbf{J}},\mathbf{J}|\mathbf{e},\phi,X_{\mathbf{T}'})$$

$$\exp\left(\frac{\lambda}{n}\sum_{k=1}^{K}\sum_{m=1}^{n} l(X_{\mathbf{T}_{0,m}}, g_\theta(e_k))\left(\mathbb{1}_{k=\bar{J}_m} - \mathbb{1}_{k=J_m}\right)\right)$$

$$\leq \log \mathbb{E}_{P(\mathbf{T})q(\mathbf{e},\theta,\phi|X_{\mathbf{T}_1})} \underset{P(\mathbf{T}')}{\mathbb{E}} q(\bar{\mathbf{J}},\mathbf{J}|\mathbf{e},\phi,X_{\mathbf{T}'})$$

$$\mathbb{E}_{\prod_{m=1}^{n}\prod_{j=0,1}P(\mathbf{T}''_{j,m})}\exp\left(\frac{\lambda}{n}\sum_{k=1}^{K}\sum_{m=1}^{n} l(X_{\mathbf{T}_{0,m}}, g_\theta(e_k))\left(\mathbb{1}_{k=J_{\mathbf{T}''_{0,m}}} - \mathbb{1}_{k=J_{\mathbf{T}''_{1,m}}}\right)\right)$$

$$\leq \frac{\lambda^2\Delta^2}{n}. \tag{24}$$

which is estimated similarly to Eq. (13). Note that there are $2n$ variables so the calculation of the upper bound is $(2\Delta\lambda/n)^2/8 \times 2n = \lambda^2\Delta^2/n$.

Next we focus on the third and fourth terms in Eq. (23). Similarly to Eq. (16), we have

$$\mathbb{E}_{X,\mathbf{T}} \sum_{k=1}^{K} \frac{1}{n} \sum_{m=1}^{n} \mathbb{E}_{q(J_m|\mathbf{e},\phi,X_{\mathbf{T}_{1,m}})q(\mathbf{e},\phi,\theta|X_{\mathbf{T}_1})} \|X_{\mathbf{T}_{0,m}} - g_\theta(e_k)\|^2 \mathbb{1}_{k=J_m}$$

$$- \mathbb{E}_{X,\mathbf{T}} \sum_{k=1}^{K} \frac{1}{n} \sum_{m=1}^{n} \mathbb{E}_{q(J_m|\mathbf{e},\phi,X_{\mathbf{T}_{1,m}})q(\mathbf{e},\phi,\theta|X_{\mathbf{T}_1})} \|X_{\mathbf{T}_{1,m}} - g_\theta(e_k)\|^2 \mathbb{1}_{k=J_m}$$

$$= \mathbb{E}_{X,\mathbf{T}} \frac{2}{n} \sum_{m=1}^{n} \left( X_{\mathbf{T}_{1,m}} - X_{\mathbf{T}_{0,m}} \right) \cdot \mathbb{E}_{q(J_m|\mathbf{e},\phi,X_{\mathbf{T}_{1,m}})q(\mathbf{e},\phi,\theta|X_{\mathbf{T}_1})} \sum_{k=1}^{K} g_\theta(e_k) \mathbb{1}_{k=J_m}$$

$$\leq \mathbb{E}_X \frac{1}{\lambda} \mathrm{KL}(\mathbf{Q}|\mathbf{P}) + \mathbb{E}_X \frac{1}{\lambda} \log \mathbb{E}_{\mathbf{P}} \exp\left( \frac{2\lambda}{n} \sum_{m=1}^{n} \left( X_{\mathbf{T}_{1,m}} - X_{\mathbf{T}_{0,m}} \right) \cdot \sum_{k=1}^{K} g_\theta(e_k) \mathbb{1}_{k=J_m} \right)$$

$$\leq \mathbb{E}_X \frac{1}{\lambda} \mathrm{KL}(\mathbf{Q}|\mathbf{P})$$

$$+ \mathbb{E}_X \frac{1}{\lambda} \log \mathbb{E}_{P(\mathbf{T})q(\mathbf{e},\theta,\phi|X_{\mathbf{T}_1})} \underset{P(\mathbf{T}')}{\mathbb{E}} q(\bar{\mathbf{J}},\mathbf{J}|\mathbf{e},\phi,X_{\mathbf{T}'}) \mathbb{E}_{\prod_{m=1}^{n} \prod_{j=0,1} P(\mathbf{T}''_{j,m})}$$

$$\exp\left( \frac{2\lambda}{n} \sum_{m=1}^{n} \left( X_{\mathbf{T}_{1,m}} - X_{\mathbf{T}_{0,m}} \right) \cdot \sum_{k=1}^{K} g_\theta(e_k) \mathbb{1}_{k=J_m} \right). \tag{25}$$

We first evaluate the expectation of the exponential moment;

$$\Omega := \mathbb{E}_{P(\mathbf{T})q(\mathbf{e},\theta,\phi|X_{\mathbf{T}_1})} \frac{2}{n} \sum_{m=1}^{n} \left( X_{\mathbf{T}_{1,m}} - X_{\mathbf{T}_{0,m}} \right) \cdot \underset{\underset{P(\mathbf{T}')}{\mathbb{E}}}{\mathbb{E}} q(\bar{\mathbf{J}},\mathbf{J}|\mathbf{e},\phi,X_{\mathbf{T}'}) \sum_{k=1}^{K} g_\theta(e_k) \mathbb{1}_{k=J_m}. \tag{26}$$

Let us now focus on the expectation $\underset{P(\mathbf{T}')}{\mathbb{E}} q(\bar{\mathbf{J}},\mathbf{J}|\mathbf{e},\phi,X_{\mathbf{T}'})$. Due to the permutation symmetry, $\underset{\underset{P(\mathbf{T}')}{\mathbb{E}}}{\mathbb{E}} q(\bar{\mathbf{J}},\mathbf{J}|\mathbf{e},\phi,X_{\mathbf{T}'}) \sum_{k=1}^{K} g_\theta(e_k) \mathbb{1}_{k=J_m}$ is the same for all $m$.

For instance, when $n = 2$, the possible permutations of $\mathbf{T}$ are $\mathbf{T} = (1,2,3,4), (1,2,4,3), (1,3,2,4), \ldots$, resulting in 24 distinct patterns and thus

$$P_{k,1} = \underset{\underset{P(\mathbf{T}')}{\mathbb{E}}}{\mathbb{E}} q(\bar{\mathbf{J}},\mathbf{J}|\mathbf{e},\phi,X_{\mathbf{T}'}) \mathbb{1}_{k=\bar{J}_1} = \mathbb{E}_{\frac{1}{4}q(J_1|\mathbf{e},\phi,X_1)+\frac{1}{4}q(J_1|\mathbf{e},\phi,X_2)+\frac{1}{4}q(J_1|\mathbf{e},\phi,X_3)+\frac{1}{4}q(J_1|\mathbf{e},\phi,X_4)} \mathbb{1}_{k=J_1}$$

$$P_{k,2} = \underset{\underset{P(\mathbf{T}')}{\mathbb{E}}}{\mathbb{E}} q(\bar{\mathbf{J}},\mathbf{J}|\mathbf{e},\phi,X_{\mathbf{T}'}) \mathbb{1}_{k=\bar{J}_2} = \mathbb{E}_{\frac{1}{4}q(J_2|\mathbf{e},\phi,X_1)+\frac{1}{4}q(J_2|\mathbf{e},\phi,X_2)+\frac{1}{4}q(J_2|\mathbf{e},\phi,X_3)+\frac{1}{4}q(J_2|\mathbf{e},\phi,X_4)} \mathbb{1}_{k=J_2}$$

$$\vdots$$

Thus, all $P_{k,m}$ does not depend on the index $m$. So we express $\underset{\underset{P(\mathbf{T}')}{\mathbb{E}}}{\mathbb{E}} q(\bar{\mathbf{J}},\mathbf{J}|\mathbf{e},\phi,X_{\mathbf{T}'}) \sum_{k=1}^{K} g_\theta(e_k) \mathbb{1}_{k=J_m}$ as $P_k$. Then Eq. (26) can be written as

$$\mathbb{E}_X \mathbb{E}_{P(\mathbf{T})q(\mathbf{e},\theta,\phi|X_{\mathbf{T}_1})} \left( \frac{1}{n} \sum_{m=1}^{n} X_{\mathbf{T}_{1,m}} - \frac{1}{n} \sum_{m=1}^{n} X_{\mathbf{T}_{0,m}} \right) \cdot \sum_{k=1}^{K} g_\theta(e_k) P_k$$

$$\leq \mathbb{E}_X \mathbb{E}_{P(\mathbf{T})q(\mathbf{e},\theta,\phi|X_{\mathbf{T}_1})} \left\| \frac{1}{n} \sum_{m=1}^{n} X_{\mathbf{T}_{1,m}} - \frac{1}{n} \sum_{m=1}^{n} X_{\mathbf{T}_{0,m}} \right\| \mathbb{E}_{P(\mathbf{T})q(\mathbf{e},\theta,\phi|X_{\mathbf{T}_1})} \left\| \sum_{k=1}^{K} g_\theta(e_k) P_k \right\|$$

$$\leq \mathbb{E}_X \mathbb{E}_{P(\mathbf{T})} \left\| \frac{1}{n} \sum_{m=1}^{n} X_{\mathbf{T}_{1,m}} - \frac{1}{n} \sum_{m=1}^{n} X_{\mathbf{T}_{0,m}} \right\| \sqrt{\Delta}$$

$$\leq \mathbb{E}_X \mathbb{E}_{\prod_{m=1}^{n} \prod_{j=0,1} P(\mathbf{T}''_{j,m})} \left\| \frac{1}{n} \sum_{m=1}^{n} X_{\mathbf{T}_{1,m}} - \frac{1}{n} \sum_{m=1}^{n} X_{\mathbf{T}_{0,m}} \right\| \sqrt{\Delta},$$

where we used the negative association property of the permutation distribution. We bound the above exactly same ways as Eq. (18), that is, we can upper bound the above by the variance of bounded random variable and thus, we have

$$\mathbb{E}_X \mathbb{E}_{\prod_{m=1}^n \prod_{j=0,1} P(\mathbf{T}''_{j,m})} \left\| \frac{1}{n} \sum_{m=1}^n X_{\mathbf{T}_{1,m}} - \frac{1}{n} \sum_{m=1}^n X_{\mathbf{T}_{0,m}} \right\| \leq 2\sqrt{\frac{\Delta}{4n}}.$$

Thus, we have

$$\Omega = \mathbb{E}_X \mathbb{E}_{P(\mathbf{T})q(\mathbf{e},\theta,\phi|X_{\mathbf{T}_1})} \left( \frac{2}{n} \sum_{m=1}^n X_{\mathbf{T}_{1,m}} - \frac{2}{n} \sum_{m=1}^n X_{\mathbf{T}_{0,m}} \right) \cdot \sum_{k=1}^K g_\theta(e_k) P_k \leq \frac{2\Delta}{\sqrt{n}},$$

Let us back to the evaluation of the exponential moment in Eq. (25), we will evaluate the following

$$\mathbb{E}_X \frac{1}{\lambda} \mathrm{KL}(\mathbf{Q}|\mathbf{P}) + \mathbb{E}_X \frac{1}{\lambda} \log \mathbb{E}_\mathbf{P} \exp \left( \frac{2\lambda}{n} \sum_{m=1}^n \left( X_{\mathbf{T}_{1,m}} - X_{\mathbf{T}_{0,m}} \right) \cdot \sum_{k=1}^K g_\theta(e_k) \mathbb{1}_{k=J_m} - \lambda\Omega \right) + \Omega.$$

We then evaluate this similarly to Eq. (24), which uses the negative association of the permutation distribution and McDiarmid's inequality. The the exponential moment is upper bounded by $(2\Delta\lambda/n)^2/8 \times 2n = \lambda^2\Delta^2/n$ We then obtain

$$\mathbb{E}_{X,\mathbf{T}} \sum_{k=1}^K \frac{1}{n} \sum_{m=1}^n \mathbb{E}_{q(J_m|\mathbf{e},\phi,X_{\mathbf{T}_{1,m}})q(\mathbf{e},\phi,\theta|X_{\mathbf{T}_1})} \| X_{\mathbf{T}_{1,m}} - g_\theta(e_k) \|^2 \mathbb{1}_{k=J_m}$$

$$- \mathbb{E}_{X,\mathbf{T}} \sum_{k=1}^K \frac{1}{n} \sum_{m=1}^n \mathbb{E}_{q(J_m|\mathbf{e},\phi,X_{\mathbf{T}_{0,m}})q(\mathbf{e},\phi,\theta|X_{\mathbf{T}_1})} \| X_{\mathbf{T}_{0,m}} - g_\theta(e_k) \|^2 \mathbb{1}_{k=J_m}$$

$$\leq \mathbb{E}_X \frac{1}{\lambda} \mathrm{KL}(\mathbf{Q}|\mathbf{P}) + \mathbb{E}_X \frac{1}{\lambda} \log \mathbb{E}_\mathbf{P} \exp \left( \frac{2\lambda}{n} \sum_{m=1}^n \left( X_{\mathbf{T}_{1,m}} - X_{\mathbf{T}_{0,m}} \right) \cdot \sum_{k=1}^K g_\theta(e_k) \mathbb{1}_{k=J_m} - \lambda\Omega \right) + \Omega$$

$$\leq \mathbb{E}_X \frac{1}{\lambda} \mathrm{KL}(\mathbf{Q}|\mathbf{P}) + \frac{\lambda\Delta^2}{n} + \frac{2\Delta}{\sqrt{n}}. \tag{27}$$

In conclusion, from Eqs. (24) and (27) we have

$$\mathrm{gen}(n, \mathcal{D}) \leq \mathbb{E}_X \frac{2}{\lambda} \mathrm{KL}(\mathbf{Q}|\mathbf{P}) + \frac{2\lambda\Delta^2}{n} + \frac{2\Delta}{\sqrt{n}},$$

and optimizing the $\lambda$, we have

$$\mathrm{gen}(n, \mathcal{D}) \leq 4\Delta \mathbb{E}_X \sqrt{\frac{\mathrm{KL}(\mathbf{Q}|\mathbf{P})}{n}} + \frac{2\Delta}{\sqrt{n}} = 4\Delta \sqrt{\frac{I(\overline{\mathbf{J}}, \mathbf{J}; \mathbf{T}|\mathbf{e}, \phi, X)}{n}} + \frac{2\Delta}{\sqrt{n}}.$$

# F    PROOF OF THEOREM 4

Here, we present the results for a general stochastic encoder. For fixed $\phi$ and $\mathbf{e}$, assume that for all $\mathbf{x} \in \tilde{X}$, for any $j \in [K]$, and for a fixed $\delta \in \mathbb{R}^+$, the following holds: $q(J = j|\mathbf{e}, f_\phi(x)) \leq e^{h(\delta)} q(J = j|\mathbf{e}, \hat{f}(x)))$ with $h : \mathbb{R}^+ \to \mathbb{R}^+$.

**Theorem 7.** *Assume that there exists a positive constant $\Delta_z$ such that $\sup_{z,z' \in \mathcal{Z}} \|z - z'\| < \Delta_z$. Then, when using Eq. (2) and under the same setting as Theorem 3, for any $\delta \in (0, 1]$, we have*

$$\mathrm{gen}(n, \mathcal{D}) \leq \Delta \sqrt{nh(\delta)} + 4\Delta \sqrt{\frac{\log \mathcal{N}(\delta, \mathcal{F}, 2n)}{n}} + \frac{2\Delta}{\sqrt{n}}.$$

To prove this lemma, we first replace the output of the encoder with that obtained using the $\delta$-cover of the encoder network. Since we assumed that $q(J = j|\mathbf{e}, \phi, x) = q(J = j|\mathbf{e}, f_\phi(x))$, we need to

approximate the error caused by $q(J = j|\mathbf{e}, \hat{f}(x))$ approximating $q(J = j|\mathbf{e}, \phi, x)$. To evaluate this gap, we apply the Donsker-Valadhan lemma between the two distributions

$$\mathbf{Q} := q(J|\mathbf{e}, f_\phi(X)) \prod_{m=1}^{n} q(J_m|\mathbf{e}, f_\phi(S_m)),$$

$$\mathbf{P} := q(J|\mathbf{e}, \hat{f}(X)) \prod_{m=1}^{n} q(J_m|\mathbf{e}, \hat{f}(S_m)). \tag{28}$$

Then we have

$$\text{gen}(n, \mathcal{D})$$

$$= \mathop{\mathbb{E}}_{S,X} \mathbb{E}_{q(\mathbf{e},\phi,\theta|S)} \left( \mathbb{E}_{q(J|\mathbf{e},f_\phi(X))} l(X, g_\theta(e_J)) - \frac{1}{n} \sum_{m=1}^{n} \mathbb{E}_{q(J_m|\mathbf{e},f_\phi(S_m))} l(S_m, g_\theta(e_{J_m})) \right)$$

$$\leq \mathop{\mathbb{E}}_{S,X} \mathbb{E}_{q(\mathbf{e},\phi,\theta|S)} \left( \mathbb{E}_{q(J|\mathbf{e},\hat{f}(X))} l(X, g_\theta(e_J)) - \frac{1}{n} \sum_{m=1}^{n} \mathbb{E}_{q(J_m|\mathbf{e},\hat{f}(S_m))} l(S_m, g_\theta(e_{J_m})) \right)$$

$$+ \mathop{\mathbb{E}}_{S,X} \mathbb{E}_{q(\mathbf{e},\phi,\theta|S)} \frac{1}{\lambda} \text{KL}(\mathbf{Q} \| \mathbf{P})$$

$$+ \mathop{\mathbb{E}}_{S,X} \mathbb{E}_{q(\mathbf{e},\phi,\theta|S)} \frac{1}{\lambda} \log \mathbb{E}_{\mathbf{P}} \exp \left( \lambda l(X, g_\theta(e_J)) - \frac{\lambda}{n} \sum_{m=1}^{n} l(S_m, g_\theta(e_{J_m})) \right)$$

$$\leq \mathop{\mathbb{E}}_{S,X} \mathbb{E}_{q(\mathbf{e},\phi,\theta|S)} \left( \mathbb{E}_{q(J|\mathbf{e},\hat{f}(X))} l(X, g_\theta(e_J)) - \frac{1}{n} \sum_{m=1}^{n} \mathbb{E}_{q(J_m|\mathbf{e},\hat{f}(S_m))} l(S_m, g_\theta(e_{J_m})) \right)$$

$$+ \frac{(n+1)h(\delta)}{\lambda} + \frac{\lambda \Delta^2}{2},$$

where we used the following relation

$$\text{KL}(\mathbf{Q} \| \mathbf{P}) \leq (n+1) \log e^{h(\delta)} = (n+1)h(\delta),$$

which is proved by the assumption of the stability. We also used the fact that $-\lambda\Delta \leq \lambda l(X, g_\theta(e_J)) - \frac{\lambda}{n} \sum_{m=1}^{n} l(S_m, g_\theta(e_{J_m})) \leq \lambda\Delta$ to uuper bound the exponential moment.

By optimizing $\lambda$, we have

$$\text{gen}(n, \mathcal{D})$$

$$\leq \mathop{\mathbb{E}}_{S,X} \mathbb{E}_{q(\mathbf{e},\phi,\theta|S)} \left( \mathbb{E}_{q(J|\mathbf{e},\hat{f}(X))} l(X, g_\theta(e_J)) - \frac{1}{n} \sum_{m=1}^{n} \mathbb{E}_{q(J_m|\mathbf{e},\hat{f}(S_m))} l(S_m, g_\theta(e_{J_m})) \right)$$

$$+ \Delta \sqrt{\frac{(n+1)h(\delta)}{2}}.$$

This implies that the first term corresponds to the generalization bound when using the $\delta$-cover of the encoder network. We can bound this term similarly to Theorem 3.

When applying the result of Theorem 3, we utilize the Donsker-Valadhan inequality for Eq. (22). Instead of using Eq. (28), we consider the following distributions:

$$\mathbf{Q} := q(\bar{\mathbf{J}}, \mathbf{J}|\mathbf{e}, f_\phi(X_{\mathbf{T}})) = \prod_{m=1}^{n} q(\bar{J}_m|\mathbf{e}, \hat{f}(\tilde{X}_{T_m})) \prod_{m=1}^{n} q(J_m|\mathbf{e}, \hat{f}(\tilde{X}_{T_{n+m}}))$$

$$\mathbf{P} := q(\bar{\mathbf{J}}, \mathbf{J}|\mathbf{e}, \hat{f}(X_{\mathbf{T}})) = \prod_{m=1}^{n} q(\bar{J}_m|\mathbf{e}, \hat{f}(\tilde{X}_{T_m})) \prod_{m=1}^{n} q(J_m|\mathbf{e}, \hat{f}(\tilde{X}_{T_{n+m}})).$$

From assumption, we have

$$\text{KL}(\mathbf{Q} \| \mathbf{P}) \leq 2n \log e^{h(\delta)} = 2nh(\delta).$$

Then from Eq. (22), we have

$$\text{gen}(n, \mathcal{D})$$

$$\leq \mathbb{E}_{X,\mathbf{T}} \sum_{k=1}^{K} \frac{1}{n} \sum_{m=1}^{n} \mathbb{E}_{q(\bar{J}_m|\mathbf{e}, \hat{f}(X_{\mathbf{T}_{0,m}}))q(\mathbf{e}, \phi, \theta|X_{\mathbf{T}_1})} \|X_{\mathbf{T}_{0,m}} - g_\theta(e_k)\|^2 \mathbb{1}_{k=\bar{J}_m}$$

$$- \mathbb{E}_{X,\mathbf{T}} \sum_{k=1}^{K} \frac{1}{n} \sum_{m=1}^{n} \mathbb{E}_{q(J_m|\mathbf{e}, \hat{f}(X_{\mathbf{T}_{1,m}}))q(\mathbf{e}, \phi, \theta|X_{\mathbf{T}_1})} \|X_{\mathbf{T}_{1,m}} - g_\theta(e_k)\|^2 \mathbb{1}_{k=J_m} + \Delta \sqrt{\frac{2nh(\delta)}{2}}$$

$$\leq 4\Delta \sqrt{\frac{I(\bar{\mathbf{J}}, \mathbf{J}; \mathbf{T}|\mathbf{e}, \hat{f}(X))}{n}} + \frac{2\Delta}{\sqrt{n}} + \Delta \sqrt{\frac{2nh(\delta)}{2}},$$

where

$$I(\bar{\mathbf{J}}, \mathbf{J}; \mathbf{T}|\mathbf{e}, \phi, X) = \mathbb{E}_{\tilde{X}, \mathbf{T}} \mathbb{E}_{q(\mathbf{e}, \phi|\tilde{X}_{\mathbf{T}_1})} \text{KL}(q(\tilde{\mathbf{J}}|\mathbf{e}, \hat{f}(\tilde{X})) \| \mathbb{E}_{P(\mathbf{T}')} q(\bar{\mathbf{J}}, \mathbf{J}|\mathbf{e}, \hat{f}(\tilde{X}_{\mathbf{T}'_0}), \hat{f}(\tilde{X}_{\mathbf{T}'_1}))).$$

Note that we consider the CMI for the discrete variable, it is upper bounded by the entropy (Cover & Thomas, 2012), and we have

$$I(\bar{\mathbf{J}}, \mathbf{J}; \mathbf{T}|\mathbf{e}, \hat{f}(X)) \leq H[\bar{\mathbf{J}}, \mathbf{J}|\mathbf{e}, \hat{f}(X)] \leq \log \mathcal{N}(\delta, \mathcal{F}, 2n).$$

The first inequality follows from the fact that MI is defined as the difference between the entropy and the conditional entropy, and the entropy of discrete variables is always non-negative. The second inequality arises because $\bar{\mathbf{J}}, \mathbf{J}$ are outputs of a function evaluated at $2n$ points. Thus, we considered the covering number at $2n$ points, defined as $\mathcal{N}(\delta, \mathcal{F}, n) := \sup_{x^{2n} \in \mathcal{X}^{2n}} \mathcal{N}(\delta, \mathcal{F}, x^{2n})$. Since the entropy is bounded above by the logarithm of the maximum cardinality, we obtain the second inequality.

Thus, we have

$$\text{gen}(n, \mathcal{D}) \leq 4\Delta \sqrt{\frac{\log \mathcal{N}(\delta, \mathcal{F}, 2n)}{n}} + \frac{2\Delta}{\sqrt{n}} + \Delta \sqrt{nh(\delta)}.$$

## F.1 BEHAVIOR OF EQ. (2)

Finally, we show that Eq. (2) satisfies $h(\delta) = 8\beta \Delta_z \delta$ because

$$\frac{q(J = j|\mathbf{e}, f_\phi(x))}{q(J = j|\mathbf{e}, \hat{f}(x))}$$

$$= \frac{e^{-\beta \|f_\phi(x) - e_j\|^2}}{e^{-\beta \|\hat{f}(x) - e_j\|^2}} \times \frac{\sum_{k=1}^{K} e^{-\beta \|\hat{f}(x) - e_k\|^2}}{\sum_{k=1}^{K} e^{-\beta \|f_\phi(x) - e_k\|^2}}$$

$$= e^{-\beta \|f_\phi(x) - e_j\|^2 + \beta \|\hat{f}(x) - e_j\|^2} \times \frac{\sum_{k=1}^{K} e^{\beta \|f_\phi(x) - e_k\|^2}}{\sum_{k=1}^{K} e^{\beta \|\hat{f}(x) - e_k\|^2}}$$

$$\leq e^{\beta(\hat{f}(x) - f_\phi(x)) \cdot (\hat{f}(x) + f_\phi(x)) - 2\beta e_j \cdot (\hat{f}(x) - f_\phi(x))} \times \sup_{k \in [K]} e^{-\beta \|\hat{f}(x) - e_k\|^2 + \beta \|f_\phi(x) - e_k\|^2}$$

$$\leq e^{4\beta \Delta_z \delta} \times e^{4\beta \Delta_z \delta}.$$

Thus we have $h(\delta) = 8\beta \Delta_z \delta$ and by substituting this into above Theorem, we obtain Theorem 4.

## F.2 DISCUSSION ABOUT THE METRIC ENTROPY FOR REGULARIZED MODEL

Here we discuss the upper bound of metric entropy in our setting. Since the latent variable lies in $\mathbb{R}^{d_z}$, the encoder network operates as $f_\phi : \mathbb{R}^d \to \mathbb{R}^{d_z}$, making it a multivariate function.

To evaluate the complexity of the metric entropy for such multivariate functions, the concept of Natarajan dimension with margin has been employed (Guermeur, 2007). According to Lemma 39 (and also Lemma 37 and 38), if a multivariate function has a finite Natarajan dimension with margin,

then its metric entropy scales as $\mathcal{O}(\log n)$. To explore the properties of the Natarajan dimension with margin, Guermeur (2018) demonstrated that it can be bounded by the fat-shattering dimension of each component of the original multivariate function (Lemma 10) under the certain margin assumption. Additionally, Guermeur (2017) showed in Lemma 1 that the covering number of the multivariate function can be bounded by the covering number of each of its components under the certain margin assumption. To further bound the covering number of each dimension, one can rely on the fat-shattering dimension of each function, as discussed in Lemma 3.5 of Alon et al. (1997). Thus, it is essential to bound the fat-shattering dimension in both cases. Examples of fat-shattering dimension evaluations can be found, for instance, in Bartlett & Maass (2003), which discusses neural network models, and Gottlieb et al. (2014), which addresses the fat-shattering dimension of Lipschitz function classes. If our encoder network adheres to these properties, we can bound its covering number accordingly.

In conclusion, if the log of the covering number satisfies $\mathcal{O}(\log n)$, by setting $\delta = 1/n^2$, we obtain that $\text{gen}(n, \mathcal{D}) = \mathcal{O}(\sqrt{\log n / n})$.

## G PROOFS FOR THE STATEMENTS IN SECTION 3.3

### G.1 PROOF OF EQ. (8)

Eq. (8) is derived by applying the Jensen inequality to the definition of the CMI in Eq. (4).

Similarly to Eq. (8), we have

$$
\begin{aligned}
I(\tilde{\mathbf{J}}; \mathbf{T}|\mathbf{e}, \phi, \tilde{X}) &= \mathop{\mathbb{E}}_{\tilde{X}, \mathbf{T}} \mathop{\mathbb{E}}_{q(\mathbf{e}, \phi|\tilde{X}_{\mathbf{T}_1})} \text{KL}(q(\tilde{\mathbf{J}}|\mathbf{e}, \phi, \tilde{X}) \| \mathop{\mathbb{E}}_{P(\mathbf{T}')} q(\bar{\mathbf{J}}, \mathbf{J}|\mathbf{e}, \phi, \tilde{X}_{\mathbf{T}'_0}, \tilde{X}_{\mathbf{T}'_1})) \\
&\leq \mathop{\mathbb{E}}_{\tilde{X}, \mathbf{T}, \mathbf{T}'} \mathop{\mathbb{E}}_{q(\mathbf{e}, \phi|\tilde{X}_{\mathbf{T}_1})} \text{KL}(q(\tilde{\mathbf{J}}|\mathbf{e}, \phi, \tilde{X}) \| q(\bar{\mathbf{J}}, \mathbf{J}|\mathbf{e}, \phi, \tilde{X}_{\mathbf{T}'_0}, \tilde{X}_{\mathbf{T}'_1})),
\end{aligned}
$$

which also implies the KL stability.

### G.2 PROOF OF EQ. (9)

Since for all $\mathbf{x}, \mathbf{x}' \in \mathcal{X}$ and for any $j \in [K]$, $q(J = j|\mathbf{e}, \phi, \mathbf{x}) \leq e^{\epsilon} q(J = j|\mathbf{e}, \phi, \mathbf{x}')$, we have

$$
\begin{aligned}
I(\tilde{\mathbf{J}}; U|\mathbf{e}, \phi, \tilde{X}) &\leq \mathop{\mathbb{E}}_{\tilde{X}, U, U'} \mathop{\mathbb{E}}_{q(\mathbf{e}, \phi|\tilde{X}_U)} \text{KL}(q(\bar{\mathbf{J}}, \mathbf{J}|\mathbf{e}, \phi, \tilde{X}_{\bar{U}}, \tilde{X}_U) | q(\bar{\mathbf{J}}, \mathbf{J}|\mathbf{e}, \phi, \tilde{X}_{\bar{U}'}, \tilde{X}_{U'})) \\
&\leq 2n \mathop{\mathbb{E}}_{S} \mathop{\mathbb{E}}_{q(\mathbf{e}, \phi|S)} \epsilon.
\end{aligned}
$$

Similarly, the empirical KL term can be bounded by setting the marginal distribution as the prior distribution,

$$
\begin{aligned}
\frac{\mathbb{E}_{q(\mathbf{e}, \phi, \theta|S)} \text{KL}(\mathbf{Q}|\mathbf{P})}{n} &= \frac{1}{n} \sum_{m=1}^{n} \mathbb{E}_{q(\mathbf{e}, \phi|S)} \text{KL}(q(J_m|\mathbf{e}, \phi, S_m) \| q(J_m|\mathbf{e}, \phi)) \\
&\leq \frac{1}{n} \sum_{m=1}^{n} \mathbb{E}_{q(\mathbf{e}, \phi|S)} \mathbb{E}_X \text{KL}(q(J_m|\mathbf{e}, \phi, S_m) \| q(J_m|\mathbf{e}, \phi, X)) \\
&\leq n \mathop{\mathbb{E}}_{S} \mathop{\mathbb{E}}_{q(\mathbf{e}, \phi|S)} \epsilon.
\end{aligned}
$$

Substituting these upper bounds into Eq. (3), we obtain Eq. (9).

### G.3 DISCUSSION ABOUT THE STABILITY OF EQ. (2)

The posterior distribution in Eq. (2) corresponds to the **exponential mechanism** in the privacy context (McSherry & Talwar, 2007) and McSherry & Talwar (2007) proved that Eq. (2) satisfies the

$\epsilon = 2\beta\Delta_\phi$-DP. For the completeness, we provide the proof. From Eq. (2), we have that

$$
\begin{aligned}
\frac{q(J = j|\mathbf{e}, \phi, x)}{q(J = j|\mathbf{e}, \phi, x')} &= \frac{e^{-\beta\|f_\phi(x)-e_j\|^2}}{e^{-\beta\|f_\phi(x')-e_j\|^2}} \times \frac{\sum_{k=1}^{K} e^{-\beta\|f_\phi(x')-e_k\|^2}}{\sum_{k=1}^{K} e^{-\beta\|f_\phi(x)-e_k\|^2}} \\
&= e^{-\beta\|f_\phi(x)-e_j\|^2 + \beta\|f_\phi(x')-e_j\|^2} \times \frac{\sum_{k=1}^{K} e^{\beta\|f_\phi(x)-e_k\|^2}}{\sum_{k=1}^{K} e^{\beta\|f_\phi(x')-e_k\|^2}} \\
&\leq e^{\beta\Delta_{\phi,\mathbf{e}}} \times \sup_{k\in[K]} e^{-\beta\|f_\phi(x')-e_k\|^2 + \beta\|f_\phi(x)-e_k\|^2} \\
&\leq e^{2\beta\Delta_{\phi,\mathbf{e}}}
\end{aligned}
$$

Then combining Eq. (9), we obtain the generalization error bound.

## H PROOF OF THEOREM 5

Define the distribution obtained by the training dataset as follows; conditioned on $\mathbf{e}, \phi, S$, we have

$$
\hat{\mu}_S = \frac{1}{n} \sum_{m=1}^{n} g_\theta \# q(e_{(m)}|\mathbf{e}, \phi, S_m)
$$

From the triangle inequality, we have

$$
W_2(\mathcal{D}, \hat{\mu}) \leq W_2(\mathcal{D}, \hat{\mu}_S) + W_2(\hat{\mu}_S, \hat{\mu}) \tag{29}
$$

The first term of Eq. (29) is bounded as follows;

$$
\begin{aligned}
\mathbb{E}_S \mathbb{E}_{q(\mathbf{e}, \phi, \theta|S)} W_2^2(\mathcal{D}, \hat{\mu}_S) &\leq \mathbb{E}_S \mathbb{E}_{q(\mathbf{e}, \phi, \theta|S)} \frac{1}{n} \sum_{m=1}^{n} \mathbb{E}_X \mathbb{E}_{q(e_{(m)}|\mathbf{e}, \phi, S_m)} \|x - g_\theta(e_{(m)})\|^2 \\
&= \mathbb{E}_S \mathbb{E}_{q(\mathbf{e}, \phi, \theta|S)} \frac{1}{n} \sum_{m=1}^{n} \sum_{k=1}^{K} \|x - g_\theta(e_k)\|^2 \mathbb{E}_{q(J_m|\mathbf{e}, \phi, S_m)} \mathbb{1}_{k=J_m}. \tag{30}
\end{aligned}
$$

The first inequality is obtained by the definition of the Wasserstein distance.

The expression inside the square root corresponds to the first term of Eq. (15). We can verify this by noting that Eq. (30) represents the squared error at the test data point $x$ under the prediction $e_k$, which is derived using the training dataset. Meanwhile, the first term of Eq. (15) represents this error when the test data is replaced by the supersample $\bar{U}$.

Therefore, Eq. (30) can be upper-bounded by applying Eq. (19), which serves as the upper bound for Eq. (15).

$$
\mathbb{E}_S \mathbb{E}_{q(\mathbf{e}, \phi, \theta|S)} W_2^2(\mathcal{D}, \hat{\mu}_S) \tag{31}
$$

$$
\leq \mathbb{E}_S \mathbb{E}_{q(\mathbf{e}, \phi, \theta|S)} \frac{1}{n} \sum_{m=1}^{n} \mathbb{E}_{q(e_{(m)}|\mathbf{e}, \phi, S_m)} \|S_m - g_\theta(e_{(m)})\|^2 + \frac{1}{\lambda} \text{KL}(\mathbf{Q}|\mathbf{P}) + \frac{\lambda\Delta^2}{2n} + \frac{\Delta}{\sqrt{n}},
$$

where

$$
\mathbf{Q} := q(\mathbf{e}, \phi, \theta|S) \prod_{m=1}^{n} q(J_m|\mathbf{e}, \phi, S_m) = q(\mathbf{e}, \phi, \theta|S) \prod_{m=1}^{n} q(e_{(m)}|\mathbf{e}, \phi, S_m),
$$

$$
\mathbf{P} := q(\mathbf{e}, \phi, \theta|S) \prod_{m=1}^{n} q(J_m|\mathbf{e}, \phi) = q(\mathbf{e}, \phi, \theta|S) \prod_{m=1}^{n} q(e_{(m)}|\mathbf{e}, \phi).
$$

Next, the second term of Eq. (29) is bounded as follows; We express $\prod_{m=1}^n q(e_{(m)}|\mathbf{e}, \phi) = q(e)$ for simplicity, then we have

$$\mathbb{E}_S \mathbb{E}_{q(\mathbf{e}, \phi, \theta|S)} W_2^2(\hat{\mu}_S, \hat{\mu}) \tag{32}$$

$$\leq \mathbb{E}_S \mathbb{E}_{q(\mathbf{e}, \phi, \theta|S)} \frac{1}{n} \sum_{m=1}^n \mathbb{E}_{q(e)} \mathbb{E}_{q(e_{(m)}|\mathbf{e}, \phi, S_m)} \|g_\theta(e) - g_\theta(e_{(m)})\|^2$$

$$= \mathbb{E}_S \mathbb{E}_{q(\mathbf{e}, \phi, \theta|S)} \left( \frac{1}{n} \sum_{m=1}^n \mathbb{E}_{q(e)} \|g_\theta(e)\|^2 + \mathbb{E}_{q(e_{(m)}|\mathbf{e}, \phi, S_m)} \|g_\theta(e_{(m)})\|^2 \right.$$

$$\left. - 2\mathbb{E}_{q(e)} g_\theta(e) \cdot \mathbb{E}_{q(e_{(m)}|\mathbf{e}, \phi, S_m)} g_\theta(e(m)) \right)$$

$$\leq \frac{1}{\lambda} \mathrm{KL}(\mathbf{Q}|\mathbf{P}) + \frac{\lambda \Delta^2}{2n},$$

where we used the Donsker Valadhan lemma for the first and third terms, changing the expectation from $\mathbf{Q}$ to $\mathbf{P}$.

Combining Eqs. (31) and (32), we have

$$\mathbb{E}_S \mathbb{E}_{q(\mathbf{e}, \phi, \theta|S)} W_2^2(\mathcal{D}, \hat{\mu})$$

$$\leq 2 \left( \mathbb{E}_S \mathbb{E}_{q(\mathbf{e}, \phi, \theta|S)} \frac{1}{n} \sum_{m=1}^n \mathbb{E}_{q(e_{(m)}|\mathbf{e}, \phi, S_m)} \|S_m - g_\theta(e_{(m)})\|^2 + \frac{2}{\lambda} \mathrm{KL}(\mathbf{Q}|\mathbf{P}) + \frac{\lambda \Delta^2}{n} + \frac{\Delta}{\sqrt{n}} \right).$$

Then by optimizing $\lambda$, we have

$$\mathbb{E}_S \mathbb{E}_{q(\mathbf{e}, \phi, \theta|S)} W_2^2(\mathcal{D}, \hat{\mu})$$

$$\leq \mathbb{E}_S \mathbb{E}_{q(\mathbf{e}, \phi, \theta|S)} \frac{2}{n} \sum_{m=1}^n \mathbb{E}_{q(e_{(m)}|\mathbf{e}, \phi, S_m)} \|S_m - g_\theta(e_{(m)})\|^2 + 2\Delta \sqrt{\frac{2}{n} \mathrm{KL}(\mathbf{Q}|\mathbf{P})} + \frac{2\Delta}{\sqrt{n}}.$$

# I EXPERIMENTAL SETTINGS

In this section, we summarize our experimental settings in Section 5. Our experiments were based on the Gaussian stochastically quantized VAE (SQ-VAE) model proposed by Takida et al. (2022), and were conducted by adapting the code from their GitHub [1] to suit our experimental configurations. Therefore, we first introduce the basics of (Gaussian) SQ-VAE in Sections I.1 and I.2 and finally explain our experimental settings in Section I.3.

## I.1 OVERVIEW OF SQ-VAE

The SQ-VAE is a generative model that, similar to VQ-VAE, employs a learnable codebook $\mathbf{e} = \{e_k\}_{k=1}^K \in \mathcal{Z}^K$. The objective of SQ-VAE is to learn the *stochastic decoder* $x \sim p_\theta(x|Z_q)$ using latent variables $Z_q$ to generate samples belonging to the data distribution $p_{\mathrm{data}}(x)$, where $p_\theta(x|Z_q) = \mathcal{N}(g_\theta(Z_q), \sigma^2\mathbf{I})$, $\mathcal{N}(m, \sigma\mathbf{I})$ is the Gaussian distribution with mean and equal variance parameter $\{m, \sigma^2\mathbf{I}\}$, $\sigma^2 \in \mathbb{R}_+$, and $\mathbf{I}$ is the identity matrix. Here, $Z_q$ is sampled from a prior distribution $P(Z_q)$ over the discrete latent space $\mathbf{e}^{d_z}$.

**In the main training process** of SQ-VAE, we assume $P(Z_q)$ to be an i.i.d. uniform distribution, identical to VQ-VAE, meaning each codebook element is selected with equal probability ($P(z_{q,i} = b_k) = 1/K$ for $k \in [K]$). Subsequently, a **second training stage** is conducted to learn $P(Z_q)$. Since computing the posterior $p_\theta(Z_q|x)$ exactly is intractable, we utilize an approximate posterior distribution $q_\phi(Z_q|x)$ instead.

At the encoding process, directly mapping from $x$ to the discrete $Z_q$ is challenging due to the discrete nature of $Z_q$. To overcome this issue, Takida et al. (2022) proposed to construct a stochastic encoder by introducing the following two processes:

---

[1] https://github.com/sony/sqvae/tree/main/vision

- **Stochastic Dequantization Process**: The transformation function from $Z_q$ to the auxiliary *continuous* variable, $Z$, denoted as $p_\psi(Z|Z_q)$, where $\psi$ is its parameters.
- **Stochastic Quantization Process**: The transformation from $Z$ to $Z_q$ is given by $\hat{P}_\phi(Z_q|Z) \propto p_\phi(Z|Z_q)P(Z_q)$ obtained via Bayes' theorem, which is represented as the categorical distribution $q(J|\mathbf{e}, \phi, x)$ through the softmax function as in Eq. (2).

We can obtain $\hat{Z}_q$ from a deterministic encoder $f_\phi(x)$, where we expect that $\hat{Z}_q$ is close to $Z_q$. Therefore, we can similarly define the dequantization process of $\hat{Z}_q$ as $Z|\hat{Z}_q \sim p_\psi(Z|\hat{Z}_q)$. By combining this process with the stochastic quantization process, we can establish the following *stochastic encoding* process from $x$ to $Z_q$: $\mathbb{E}_{q_\omega(Z|x)}[\hat{P}_\phi(Z_q|Z)]$, where $\omega := \{\phi, \psi\}$ and $q_\omega(Z|x) :=$ $p_\psi(Z|f_\phi(x))$.

According to these facts, we can derive the following evidence lower bound (ELBO) for SQ-VAE:

$$-\mathcal{L}_{\mathrm{SQ}}(x; \theta, \omega, \mathbf{e})$$

$$:= \underbrace{\mathbb{E}_{q_\omega(Z|x), \hat{P}_\phi(Z_q|Z)}\left[\log\frac{p_\theta(z|Z_q)p_\phi(Z|Z_q)}{q_\omega(Z|x)}\right]}_{=\mathrm{KL}(\mathbf{Q}\|\mathbf{P})} + \mathbb{E}_{q_\omega(Z|x)}H(\hat{P}_\phi(Z_q|Z)) + (\mathrm{Const.}),$$

where $H(\hat{P}_\phi(Z_q|Z))$ is the entropy of $\hat{P}_\phi(Z_q|Z)$.

From the above, the optimization problem of SQ-VAE is minimizing $\mathbb{E}_{p_{\mathrm{data}}(x)}[\mathcal{L}_{\mathrm{SQ}}(x; \theta, \omega, \mathbf{e})]$ w.r.t. $\{\theta, \omega, \mathbf{e}\}$. This approach eliminates the need for heuristic techniques traditionally required, such as stop-gradient, exponential moving average (EMA), and codebook reset (Williams et al., 2020).

Moreover, the categorical posterior distribution $\hat{P}_\phi(Z_q|Z) = q(J|\mathbf{e}, \phi, x)$ can be approximated using the Gumbel–Softmax relaxation (Jang et al., 2017; Maddison et al., 2017), where the Gumbel–Softmax function is defined as, for all $k$ $(1 \le k \le K)$,

$$\frac{\exp(-\beta\|f_\phi(x) - e_k\|^2 + G_k)/\tau)}{\sum_{j=1}^{K}\exp(-\beta\|f_\phi(x) - e_j\|^2 + G_j)/\tau)},$$

where $G_k$ is an i.i.d. sample from the Gumbel distribution and $\tau$ is the temperature parameter that is deferent from $\beta$ in Eq. (2). This allows the application of the reparameterization trick from VAEs during backpropagation, enabling efficient gradient computation and model training.

## I.2 GAUSSIAN SQ-VAE

Gaussian SQ-VAE assumes that the dequantization process $p_\psi(Z|Z_q)$ follows a Gaussian distribution. In this paper, we set the following Gaussian distribution: $p_\psi(Z_i|Z_q) = \mathcal{N}(Z_{q,i}, \sigma_\psi^2\mathbf{I})$, where $\sigma_\psi^2 \in \mathbb{R}_+$. Then, the stochastic decoder and the stochastic dequantization process in SQ-VAE can be written as $p_\theta(x|Z_q) = \mathcal{N}(g_\theta(Z_q), \sigma^2\mathbf{I})$ and $p_\psi(Z_i|\hat{Z}_q) = \mathcal{N}(\hat{Z}_{q,i}, \sigma_\psi^2\mathbf{I})$.

## I.3 DETAILS OF EXPERIMENTAL SETTINGS

**Dataset:** We used the MNIST dataset (LeCun et al., 1989), which is $28 \times 28$ gray scale images with 10 classes. We prepared the subset dataset with $\{1000, 2000, 4000, 8000\}$ samples from the default training dataset (60000 samples). Then, we split it as the training and the validation datasets following the supersample setting as in Section 2.2.

**Model architecture and training procedure:** We adopted the ConvResNets with the architecture provided by Google DeepMind [2]. We summarize the details of this model in Table 1.

Regarding the training procedure, we adopted the settings in Takida et al. (2022) as follows. We used the Adam optimizer with 0.001 initial learning rate. The learning rate was halved every 3 epochs if the validation loss is not improving. We trained the model 200 epochs with 32 mini-batch size. As for the annealing schedule for the temperature parameter of the Gumbel-softmax sampling, we set $\tau = \exp(10^{-5} \cdot t)$ as in Jang et al. (2017), where $t$ is the global training step size.

---

[2] `https://github.com/deepmind/sonnet/blob/v2/examples/vqvae_example.ipynb`

Table 1: Experimental settings on MNIST.

| Experimental setup for MNIST experiments | |
|---|---|
| Model | Gaussian stochastically quantized VAE (SQ-VAE) (Takida et al., 2022) |
| Network archtecture | ConvResNets with three convolutional layers, two transpose convolutional layers, and one ResBlocks. |
| The size of a codebook ($K$) and the dimension of the latent space $d_z$ | $K = \{16, 32, 64, 128\}$; $d_z = 64$ |
| Optimizer | Adam with 0.001 initial learning rate |
| Batch size | 32 |
| Num. of training/validation samples | [250, 1000, 2000, 4000] |
| Num. of epochs | 200 |
| Num. of samples for CMI estimation | 3 |
| Num. of samplings for $U$ | 5 |

**GPU environment:** We used NVIDIA GPUs with 32GB memory (NVIDIA DGX-1 with Tesla V100 and DGX-2) in our experiments.

**Mutual information estimation:** To estimate the mutual information $I(\tilde{\mathbf{J}}; U|\mathbf{e}, \phi, \tilde{X})$ in Eq. (3), we developed a plug-in estimator for it, which is computed using estimators for the probability density of $\tilde{\mathbf{J}}$ and $\tilde{X}$, as well as their joint probability density, employing $k$-nearest-neighbor-based density estimation (Loftsgaarden & Quesenberry, 1965). The estimation strategy is incorporated into the `sklearn.feature_selection.mutual_info_classif` function [3]. We set $k = 3$ following the default setting of this function and Kraskov et al. (2004); Ross (2014).

---

[3]`https://scikit-learn.org/stable/modules/generated/sklearn.feature_selection.mutual_info_classif.html`

