# OpenReview forum: "Information-theoretic Generalization Analysis for Vector-Quantized VAEs"
_ICLR.cc/2025/Conference — Submitted to ICLR 2025_

### Official Review · Reviewer_c5RX · 2024-10-23

**Soundness:** 3
**Presentation:** 3
**Contribution:** 2
**Rating:** 5
**Confidence:** 2

**Summary:**

This work aims to provide insight on the generalization property of encoder-decoder models in unsupervised settings. In particular, the authors focus on encoder-decoder models with discrete latent space, e.g. Vector-quantized VAEs. For such specific architecture, the authors show theoretically that the generalization gap can be evaluated with the mutual information and that, surprisingly, it does not depend on the decoder's parameter. This can be traced back to the use of the permutation symmetric super-sample setting, which makes the empirical KL term vanish.

**Strengths:**

- the result that the generalization bound does not depend on the decoder's parameter is indeed interesting

**Weaknesses:**

- The structure of the results and most of the proof techniques are similar to [1]. The novel part comes from considering encoder-decoder models with discrete latent space and the permutation symmetric setting, which is also limiting its applicability.

[1] Mbacke et al., Statistical Guarantees for Variational Autoencoders using PAC-Bayesian Theory, NeurIPS 2023

**Questions:**

Since the topic of this paper is not exactly my area of expertise, I gave a low confidence (2) to my judgment and I am happy to read and take into consideration what other reviewers opinions.

I would like to make some comments and questions:
- could the authors clarify on why generalization bounds that depend on the encoder parameter would obfuscate the role of latent variables? (line 98)
- I would make clear in the abstract that the theoretical results of the paper are limited to encoder-decoder architectures with discrete latent variables, and not any (continuous) latent variables. Currently, this is mentioned only in the title with "Vector-quantized VAEs", but I believe it should be stressed in the abstract as well (e.g. in lines 17-18-19).
- Would the authors consider using the terminology "discrete latent variables" (as in the original VQ-VAEs paper) instead of "finite latent variables"? (e.g. line 106) I think it would help improve the scope of the paper.

Minor typos:
- line 152 "fo" should be "of"
- line 152-152 I think the sentence should be re-written
- line 330 "where as" should be "whereas"

---

> ### Author Response · Authors · 2024-11-22
> **Author response**
>
> We thank you for your feedback.
>
> ## Q.1 Regarding the clarification of ``obscuring the role of latent variables’’ in line 98
> A. In response to this concern, we kindly refer you to our explanation provided in our response to all reviewers as well as in our response to Reviewer FwXK’s Q.3
> We would also like to inform you that these discussions have been added to the revised manuscript (Appendix F.3).
>
> ## Q.2 Clarification of the limitation in the abstract
> A. As you pointed out, we have explicitly addressed the limitations in the Abstract part.
>
> ## Q.3 Regarding the expression of “finite latent variables”
> A. We thank the reviewer for this suggestion and agree that the proposed terminology is more appropriate. We have revised the paper accordingly.
>
> ## Q.4 Regarding the typos
> A. Thank you for your careful review. We have corrected the typos you pointed out.

---

> > ### Author Response · Authors · 2024-11-30
> > **Any discussion?**
> >
> > Dear Reviewer c5RX,
> >
> > Thank you very much for taking the time to review our paper.
> >
> > We hope that our responses have addressed your concerns satisfactorily.
> > As the discussion period is nearing its end, we kindly ask you to let us know if there are any remaining points that require further discussion. If our responses have resolved your concerns, we would be grateful if you could consider revisiting your score.
> >
> > Thank you for your time and consideration.
> >
> > Sincerely,
> > --Authors

---

### Official Review · Reviewer_6byK · 2024-11-04

**Soundness:** 3
**Presentation:** 3
**Contribution:** 2
**Rating:** 6
**Confidence:** 3

**Summary:**

In this article, the authors provide various bounds on the generalization and generative capabilities of encoder-decoder models, and support some of their findings with numerical experiments. Specifically, the study focuses on encoder-decoder models in an unsupervised setting with a discrete latent space. The authors present novel bounds that emphasize the role of latent variables.

Using what is known as the "supersample setting", the authors derive an upper bound on the generalization error that does not depend on information from the decoder. Additionally, by introducing a new setting, referred to as the "permutation symmetry (supersample) setting", the authors establish a new upper bound that vanishes as the number of samples increases, provided appropriate regularization is applied. This bound underscores the benefits of regularization in these models.

Their final bound addresses the generative capability of the model: the 2-Wasserstein distance between the true and generated distributions is shown to be upper-bounded. Notably, the authors' analysis includes the fact that the models are trained. Lastly, the authors estimate the quantities appearing in their bounds on real models, demonstrating that the model behavior aligns with their theoretical bounds.

**Strengths:**

The paper is well-written, with a clear structure, precise notation, and sound results. The studied models - encoder-decoder models with discrete latent variables - are not only of theoretical interest but have also shown strong empirical performance, making them deserving of attention. This study builds on and extends previous work by incorporating the effects of training and focusing on the role of latent variables. Additionally, the authors introduce a novel setting, the "permutation symmetry supersample setting," which enables new proofs for bounds. The proofs are clearly explained in the appendix. Overall, this work represents a valuable step toward a better understanding of the theoretical capabilities of encoder-decoder models.

**Weaknesses:**

My main criticism of the work lies in the numerical experiments. The increase in log K (line 468) seems to be provided with relatively limited numerical evidence. There are only four data points, and no fitting or estimation is conducted to verify if the obtained points indeed behave as log K. Moreover, the behavior in log K could be better highlighted by appropriately adjusting the scales on the plot. Additional data points for both Figures 1 and 2, and an increased K range for Figure 2, would, in my opinion, provide more solid evidence.

As a second criticism, while the formal results appear mathematically sound, their interpretation is sometimes vague or overstated. For example, the phrase "obscuring the role of latent variables" (lines 97–98) is somewhat unclear, and the term "comprehensive" (lines 97–98) also seems exaggerated, given the limitations in the considered setting and the lack of evidence that better bounds could not exist. I would recommend clarifying that these interpretations arise from bounds and should be viewed as "hints" toward the true behavior of the generalization or generative capabilities.

This also leads me to reflect on the significance of the work: while the new bounds improve upon previous bounds and extend the setting, and the novel idea in the proof—the "permutation symmetric setting"—is valuable, the insights provided by these bounds into the workings of encoder-decoder models seem moderate. However, given my limited knowledge of the literature in this domain, I may reevaluate my opinion on the significance of this work, whether in its mathematical contributions or its interpretation, after discussion with the authors.

Some more specific comments, reflecting the points above, are provided in the next section. In particular, some parts of the appendix are lacking in clarity and seem unpolished (e.g. 1012-1019 seems unfinished).

**Questions:**

Comments, Questions, and Suggestions:
- Lines 97–98: This is discussed later in the manuscript, but it might be helpful to clarify earlier what is meant by "obscuring the role of latent variables." Why is it problematic that the bounds depend on the learned decoder parameters?-
- Line 115: This claim should be nuanced. Perhaps specify the setting and its limitations. The term "comprehensive" might be an overstatement, as there are limitations in the considered setting (e.g., finite latent variables, as mentioned on page 9).
- Line 185: It may help to provide additional details on how you translate the theorem from Hellström & Durisi (2022) to your setting, as there is no "R" term, no 1/n, and no summation.
- Lines 263–269: This part is confusing: why do you mention f_\phi just after the deterministic decoder? Did you mean the encoder? What is obtained by using Theorem 8 from Hellström & Durisi (2022)? Be more precise about the regularization required for the result to hold.
- Line 270: Do you have only numerical evidence that the empirical KL term is much larger than the CMI? Please clarify.
- Lines 274–279: What exactly do you deduce from the use of the Vamp prior in Theorem 2? Specify what you prove in Appendix C.3.
- Line 312: Similar to before, why is f_phi referred to as the decoder?
- Line 330: "whereas the decoder's capacity does not affect the generalization gap" — can you confirm this applies only to this specific bound? In other settings, the decoder's capacity could impact the generalization gap. If so, consider nuancing this sentence.
- Lines 340 and 347: Perhaps clarify that the theorem does not provide quantitative information on the amount of improvement.
- Line 352: Could you clarify what is meant by the "stability of KL divergence under different data points"?
- Line 440: What does "highlighting [...] of the latent variables as the regularization" mean? Which regularization is being referred to?
- Lines 453–454: "the excellent generalization performance" — Please specify a point of comparison; excellent compared to what? Additionally, avoid using "excellent" here.
- Line 455: "Elucidate" may be too strong a term.
- Line 464: Could you explore using a prior other than the uniform distribution? Is it feasible to implement a prior in the supersample setting? If not, clarify this here. The discussion on line 482 might fit better here.
- Line 467: If the claim "This result indicates that our bound increases only by log K " is based solely on Figure 2, it seems to have relatively limited evidence. Either significantly nuance this claim or generate additional data (e.g., for a larger K and a plot that better highlights the log behavior).
- Line 474: "first comprehensive analysis" — As in the introduction, "comprehensive" may be overstated. Specify here that the study focuses on quantized encoder-decoder models rather than general encoder-decoder models. Briefly mention the bounds provided by your analysis.
- Line 778: Consider providing more detail on how you use the sub-Gaussian property.
- Line 798: Could you clarify why the inequality in line 795 demonstrates that the decoder information cannot be eliminated from the naive IT bound? From what I understand, we have gen<=Delta \sqrt(2/n ... <= I(theta...)+I(e_J...), but the fact that I(theta…) appears here does not necessarily show that it influences the naive IT bound.
- Figure 1: Please add a note in the legend about where to find definitions of the different terms.
- Figures 1 and 2: Please specify what the gray and red areas represent (I assume these are standard deviations). Is there no standard deviation for the CMI/n vs. n curves?
- Figures 1 and 2: Are there specific reasons why there are only four points per graph?
- Lines 1012–1019: This section seems incomplete.
- Appendix D1 Proof: It could be helpful to divide the proof into steps with more explanation.
- Line 1088: "Finally, we remark the CMI of Eq. (7)." — This phrase is unclear; please clarify.
- Line 1662: Is there a reason for not using more samples?
- Line 1663: The dataset splitting method is unclear. Section 2.2 does not mention validation datasets. How do you transition from {1000, 2000, 4000, 8000} to [250,1000, 2000, 4000]?
- Line 1670: Did you observe that the training loss stopped decreasing after 200 epochs? What is the rationale for choosing this number of iterations?
- Line 1688: Could you provide more details on how mutual information was estimated? This paragraph lacks specificity.

Typos and Suggested Reading Improvements (No response needed for these; ignore if irrelevant):
- Lines 38 and 607: Grnwald → Grünwald
- Line 145: It might help to specify that "randomized algorithm" is defined in Section 2.2, as this could confuse readers otherwise.
- Line 152: Typo: ", fo" → ". For"
- Lines 174–175: Clarify "associated" in this context; it may be clearer to define only U in this sentence, and its purpose becomes obvious just by continuing reading.
- Line 231: Typo: "information;" → "information."
- Line 238: Typo: missing second "|" in KL term.
- Line 247: Rephrase as "Since [...] are equivalent, we can express the first term [...]"
- Line 330: Typo: "deconder" → "decoder"
- Line 350: Typo: missing "|" in KL divergence term.
- Line 354: Typo: ";" should likely be ":".
- Line 446: Typo: "Of our provided in" → unclear phrasing.
- Line 453: Typo: "relaxation Jang et al. (2017), ..." → "relaxation (Jang et al., 2017, ...)"
- Line 453: Typo: "the excellent" → "an excellent"
- Line 561: Is the URL necessary here?
- Line 607: Verify the second author’s name.
- Line 632: Typo: "Natarajan" should be capitalized.
- Standardize author name formatting in the bibliography (either use full names or initials consistently).
- Line 772: Possible typo: X should consistently use a tilde in this line and the next.
- Line 772: Possible typo: J_mm  should have a bar at the end.
- Line 810: Should this be "generalization gap"?
- Line 830: Should it be J_n instead of j_n
​-  Equation 11: Indicate here that the 2nd and 3rd lines yield zero if that’s how the term was obtained.
- Line 863: Typo: only one "|" in KL.
- Line 882: Define "coefficients of stability" in Lemma 1.
- Line 1007: Consider clarifying "by optimizing λ".
- Line 1182: Typo: "then" appears twice.
- Line 1282: Typo: "does" → "do".
- Line 1265: Typo: ";" should likely be ":".
- Line 1314: Typo: "The the".
- Line 1382: Typo: "uuper bound".
- Line 1450: Specify which theorem.
- Line 1566: Typo: ";" should likely be ":".
- Line 1597: Typo: "settings in" → "settings of".
- Line 1608: Typo: should be sigma squared in second mention of normal distribution

---

> ### Author Response · Authors · 2024-11-22
> **Author response**
>
> We would like to show our appreciation for your insightful feedback.
>
> ## Q.1 Regarding the behavior of $\mathcal{O}(\log K)$ in Figure 2.
> A. First, we have added a straight line with a slope of $\mathcal{O}(\log K)$ to the middle plot of Figure 2., along with the results of fitting it to the CMI values measured by the coefficient of determination ($R^2$) and mean squared error (MSE). Following Reviewer iw6f’s suggestion, we have also conducted experiments on CIFAR-10, and the corresponding results are presented below.
>
> - MNIST: https://ibb.co/HrB6nFq
> - CIFAR10: https://ibb.co/Xs663FL
>
> From these results, we observe that the $\mathcal{O}(\log K)$ line achieves $R^2=1$ and an MSE of $0.000$ in both experimental settings. Based on this observation, the behavior of $O(\log K)$ is sufficiently established, and increasing $K$ is unlikely to change the results.
> Similar behavior is observed on MNIST and a more complex dataset (CIFAR-10), further supporting the promising interpretation of the CMI’s behavior.
>
> Nevertheless, demonstrating this empirically is an intriguing direction, and we appreciate your insightful comment. Unfortunately, we can anticipate that we cannot provide additional results during the rebuttal period due to the significant increase in computational time required for larger values of $n$ or $K$. However, we plan to include results with extended settings for $n$ and $K$ in the new Appendix section (Appendix J).
>
> ## Q.2 Regarding some unclear explanations in lines 97–98.
> A. We apologize for any unclear or misleading expressions in our original manuscript.
> First, we used the term “ambiguous” to describe the uncertainty about whether the encoder or the decoder in VQ-VAE is fundamentally related to generalization performance.
> As mentioned in line 205, when using existing IT bounds (see Appendix B), which depend on the complexities of both the encoder and decoder, it becomes unclear whether the generalization of VAE models is due to meaningful representations extracted by the encoder or simply the use of a complex decoder. For further details, please refer to our response to all reviewers.
> In this context, our bound explicitly demonstrates that the encoder determines the generalization capability of VQ-VAE, which is why we used the term “comprehensive.”
>
> We have clarified this discussion and added it to Section 1.2 and Appendix F.3 in the revised manuscript.
>
> ## Q.3 Regarding the significance of our new proof idea
> A. For details on how our bound improves upon existing bounds, we kindly refer you to our response to all reviewers.
> Additionally, the theoretical insights obtained from existing studies and the novel findings revealed in our work are thoroughly explained in our response to Reviewer FwXK’s Q.3.
> We hope that these responses adequately address your concerns.
>
> In this response, we focus on explaining the novelty of our proposed proof technique.
>
> Existing supersample-based analytical approaches intuitively leverage the symmetry between test and training data samples. However, as discussed in Appendix C.3, such symmetry does not hold for latent variable models like VAEs, rendering existing analytical methods inapplicable.
>
> To address this issue, we devised a new analytical approach that considers the order symmetry of training and test samples. This concept represents the most innovative and significant aspect of our analytical idea.
>
> In the context of latent variable models, the order symmetry of samples is generally known to hold owing to de Finetti's theorem, supporting the notion that our analytical approach is not based on strong and unrealistic assumptions but is instead a natural method for analyzing latent variable models such as VQ-VAE.
>
> ## Q.4 Regarding the proposal for revising the paper
> A. Thank you very much for reading our paper so carefully. We found all of your suggestions to be valid, and we have incorporated them into the revised manuscript.

---

> ### Comment · Reviewer_6byK · 2024-11-26
> **Some additional questions**
>
> Thank you for your detailed response and sorry for my late reply. I still have a few points that I am not sure to understand:
> - There is probably something trivial I am missing, but can you provide additional details on how you plot your straight line with slope O(log(K)) ? The x-axis scaling is hard to see from your values, but it seems to be linear, is this correct ?
> - What about figure 2 right ? Can you comment on it ?
> - Can you clarify if the log K behavior is only obtained from your numerics ?
> - Can you comment on this remark: Line 798: Could you clarify why the inequality in line 795 demonstrates that the decoder information cannot be eliminated from the naive IT bound? From what I understand, we have gen<=Delta \sqrt(2/n ... <= I(theta...)+I(e_J...), but the fact that I(theta…) appears here does not necessarily show that it influences the naive IT bound.
> - Can you comment on this remark: Line 270: Do you have only numerical evidence that the empirical KL term is much larger than the CMI? Please clarify.
>
> Thank you in advance for your clarifications.

---

> > ### Author Response · Authors · 2024-11-30
> > **Reply from authors**
> >
> > Thank you for your additional questions, and we apologize for the delay in our response.
> > Here are our answers to your questions:
> >
> > Q.1: Regarding the way to plot $\mathcal{O}(log K)$ and the x-axis scaling
> > A.: To illustrate $\mathcal{O}(\log K)$, we plotted the values of $c \cdot \log K$ using a constant $c > 0$. To verify whether the numerical values from our bound ( $\mathrm{CMI}/n$ ) follow $\mathcal{O}(\log K)$,  $c$ was adjusted so that the scale of $c \cdot \log K$ aligns with that of  $\mathrm{CMI}\_{mean}=\mathrm{CMI}/n$ . Specifically, $c$ was computed as $c = \mathrm{mean}(\mathrm{CMI}\_{means}/K)$, which captures the scale difference between $\mathrm{CMI}\_{mean}$  and  $K$ in average. Here,  $\mathrm{cmi\_means}$ represents the average values from four experimental settings:  $\mathrm{CMI}\_{means} = [\mathrm{CMI}\_{mean,1}, \mathrm{CMI}\_{mean,2}, \mathrm{CMI}\_{mean,3}, \mathrm{CMI}\_{mean,4}]$. This approach determines a scaling factor $c$ that best fits the observed data to the theoretical model on average.
> >
> > For the x-axis, the values of  $K$  are arranged linearly as  $K = [2^4, 2^5, 2^6, 2^7]$. If  $\mathrm{CMI}/n$ follows $\mathcal{O}(\log K)$, the theoretical values align with  $\log K = [4 \cdot \log 2, 5 \cdot \log 2, 6 \cdot \log 2, 7 \cdot \log 2]$, forming a straight line when $\log 2$  is treated as constant. In our figure, the observed behavior matches this theoretical trend, confirming that the CMI values are consistent with $O(\log K)$.
> >
> > Q.2 Regarding Figure 2 (right-hand side)
> > A. The rightmost plot in Figure 2 illustrates the changes in KL complexity as the model complexity varies ( $K = \{16, 32, 64, 128\}$ ). While $\mathrm{CMI}/n$ scales as $\mathcal{O}(\log K)$ and remains as small as $\mathcal{O}(10^{-3})$,  $\mathrm{KL}/n$ reaches magnitudes as large as $\mathcal{O}(10^{2})$, although its slower growth rate compared to $\mathrm{CMI}/n$. This indicates that $\mathrm{KL}/n$ leads to vacuous bounds as model complexity increases.
> >
> > We will added this explanation in our experimental section.
> >
> > Q.3: Regarding the theoretical behavior of $\mathcal{O}(\log K)$
> > A. The behavior of $\mathcal{O}(\log K)$ can be theoretically justified. In our setting, as shown in Eq. (20) on line 1076, it can be proven that the behavior follows $\mathcal{O}(\log K)$ if the encoder network satisfies the specified conditions. The Natarajan dimension, a generalization of the VC dimension to multiclass settings, is known to hold for parametric neural networks, as established in [Ying, 2023]. Therefore, the conditions in our setting can be satisfied.
> >
> > [Ying, 2023]: Ying Jin. Upper bounds on the natarajan dimensions of some function classes. In 2023 IEEE International Symposium on Information Theory (ISIT), pp. 1020–1025. IEEE, 2023.
> >
> > Q.4: Regarding our statement of Line 798.
> > A.: For example, when examining the naive but tight upper bound in Theorem 6, it is evident that CMI necessarily incorporates information about the decoder $g_\theta$. This is because, without leveraging the decoder’s information, it is unclear how to evaluate the extent of the loss. Consequently, in the naive bound, the information about $\theta$ is indispensable.
> >
> > For instance, further evaluating the order of the naive bound in Theorem 6 using CMI in conjunction with metric entropy requires specifying the function class of the decoder, which explicitly highlights this dependency.
> >
> > In contrast, the upper bound on line 795 is included to illustrate that even if the upper bound in Theorem 6 is transformed using standard information-theoretic techniques, such as the data processing inequality, it remains impossible to eliminate the decoder’s information from the bound.
> >
> > Q.5: Regarding the statement of Line 270.
> > A. It can be confirmed from the definition in Eq. (4) that KL is larger than CMI. By its definition, CMI can be expressed in terms of KL. In general, mutual information can be written as $MI(J; U) = \mathbb{E}_U[KL(q(J|U) \| q(J))]$, which represents the KL divergence between the conditional distribution and the marginal distribution (various conditionals are omitted here for simplicity).
> >
> > If the marginal distribution is replaced by an arbitrary distribution  p(J) , the following relationship holds: $MI(J; U) = \mathbb{E}_U[KL(q(J|U) \| q(J))] = \mathbb{E}_U[KL(q(J|U) \| p(J))] - KL(q(J) \| p(J)) \leq \mathbb{E}_U[KL(q(J|U) \| p(J))]$.
> >
> > This shows that the empirical KL bounds the marginal mutual information when $p(J)$ is interpreted as the prior distribution. In this case, since the prior distribution is uniform and significantly different from the true marginal distribution, we believe this explains the large gap observed.
> >
> > The details of these relationships are thoroughly explained in Eq. (5) and its proof in Appendix D.1.

---

> > > ### Comment · Reviewer_6byK · 2024-12-01
> > >
> > > Thank you for your detailed response, which has helped clarify my doubts. While the claims regarding the log(K) behavior were convincingly supported in the rebuttal, my perspective on the practical significance of the work remains unchanged. As such, I will maintain my current score. That said, I would like to emphasize that I find the work valuable, particularly for the novel proof technique introduced.

---

> > > > ### Author Response · Authors · 2024-12-01
> > > > **Reply from authors**
> > > >
> > > > We sincerely appreciate you taking the time to carefully read our responses despite your busy schedule.
> > > > Additionally, we are truly pleased that you have understood and recognized the value of our research.
> > > >
> > > > While there may be some repetition in our explanations, we have provided additional explanation regarding the practicality of our approach in our responses to Reviewer FwXK. We would be grateful if you could review them as well.
> > > >
> > > > We hope this clarifies that the goal of our research is to contribute to the advancement of ‘practical generalization performance evaluation for VQ-VAE.'
> > > > Thank you once again for your invaluable feedback and constructive discussion.

---

### Official Review · Reviewer_iw6f · 2024-11-04

**Soundness:** 2
**Presentation:** 2
**Contribution:** 2
**Rating:** 6
**Confidence:** 3

**Summary:**

The authors present an information-theoretic generalization analysis for VQ-VAEs, focusing on the role of latent variables. The authors develop a novel generalization error bound using IT analysis, which evaluates the reconstruction loss without using the decoder. They also introduce a permutation-symmetric supersample setting, which in turn allows for controlling the generalization gap by regularizing the encoder's capacity. In addition, the authors provide an upper bound on the 2-Wasserstein distance between the true and generated data distributions, supporting insights into the model’s data generation capabilities.​

**Strengths:**

* The authors present novel theoretical bounds on the generalization of VQ-VAE. providing insights that can potentially lead to improved data generation capabilities.
* The task of deriving practical bounds for useful models such as VQ-VAE is Herculean, and when successful can have large impact. This paper can potentially be a step towards such bounds.

**Weaknesses:**

* The assumption of discrete latents and Gaussian decoder are very strong (I want to commend the authors for pointing them out), and as such are leading to interesting bounds but with limited practical use (e.g., decoder based on normalizing flows or over discrete data cannot   be used in this setting).
* The MI estimator (Kraskov) will not work for high dimensional data or data that is far away from uniformity (i.e., highly clustered). This puts strong limitation on the applicability of the method in the general case.
* A major issue with the paper is the limited experimentation setting. The experiments focus on the generalization gap but fail to show the implication in practice for the quality of the generation. That is, how a tighter bound on the generalization gap can be used in practice, for instance to predict the usefulness of the learned representations in downstream tasks, or the quality of the generation samples.
* Also, the choice of MNIST (a simple dataset that can often lead to conclusions that do not generalize to more complex data such as ImageNet) is falling short in demonstrating that the proposed analysis is practical for more complex datasets.

**Questions:**

* Did you try more complex datasets, other than MNIST?
* Can the method be used with an auto-regressive Gaussian decoder?

---

> ### Author Response · Authors · 2024-11-22
> **Author response**
>
> First of all, we appreciate your feedback.
>
> ## Q. Did you try more complex datasets, other than MNIST?
>
> A. The primary focus of this study was to reveal the theoretical properties of the generalization performance of VQ-VAE. The purpose of the experiments was to validate the analytical results, and therefore, we limited our experimental settings to MNIST.
> However, your insightful comments made us aware that readers might also be interested in experimental results on more complex datasets.
> In response to your suggestion, we conducted additional experiments on CIFAR-10, and the results are presented below (the URLs for these figures are automatically deleted after 3 months).
>
> - CIFAR10 results: https://ibb.co/Wyq9JdY
> - CIFAR10 results (vs K): https://ibb.co/JtCXxJ2
>
> The results show that, similar to the MNIST experiments, the CMI term (center plot) decreases as $n$ increases, indicating a correlation between the behavior of the generalization gap (left plot) and the CMI term. Additionally, the values of the CMI term closely align with those of the generalization gap.
> In contrast, the KL divergence term (right plot) with a uniform prior takes extremely large values, exhibits a low correlation with the generalization gap, and does not necessarily decrease as $n$ increases.
> These findings support the importance of evaluating generalization error by designing an appropriate prior based on supersamples, as discussed in Section 3.1.
>
> ## Q.2 Regarding the applicability to Autoregressive Structures
> A. In conclusion, our bound is applicable to autoregressive decoders.
> Our main results, Theorems 2 and 3, hold for any decoder architecture when the reconstruction error is measured using the squared loss.
> This fact supports the critical insight revealed in our study: generalization performance depends solely on the complexity of the encoder; that is, no matter how complex the decoder becomes (e.g., even when using an autoregressive decoder), it has no impact on the generalization performance.
>
> ## Q.3 Regarding the MI estimator for high dimensional data
> It is indeed well-known that Kraskov’s MI estimator deteriorates as the dimensionality of the data increases.
> First, please note that the MI estimation in this study is calculated based on the latent variable space, not directly on the image dataset.
> It is worth mentioning that the dimensionality of the latent variable space is typically determined by the user and, due to the nature of the encoding task, is generally set to be lower than that of the image dataset.
> Furthermore, in our numerical experiments (Figure 2 and the second experiment discussed above), we observed that non-vacuous estimates can still be obtained even when the dimensionality $K$ of the latent variable space increases to some extent.
> Moreover, these estimates were found to correlate with the behavior of the generalization error.
> This observation indicates that the MI term in our bound effectively functions as a measure of the generalization performance of VQ-VAE.

---

> > ### Comment · Reviewer_iw6f · 2024-11-22
> > **I increased the score**
> >
> > Thank you for your detailed rebuttal.
> > You addressed some of my concerns, and I increased the score of the paper.
> > Please do include the CIFAR results in the final paper.

---

### Official Review · Reviewer_FwXK · 2024-11-04

**Soundness:** 3
**Presentation:** 1
**Contribution:** 2
**Rating:** 5
**Confidence:** 3

**Summary:**

The authors study the generalization capabilities of autoencoder models using the "supersample" setting from statistical learning theory. In the supersample setting, we assume we have a dataset, and we randomly select half of the data to train our model and use the other half for testing. Then, supersample bounds aim to show that the generalization error of our models is upper bounded by the amount of information the model's test loss reveals about how we created the train-test split.

The authors prove such supersample bounds for the settings where the generalization error is measured by the model's average reconstruction loss and the 2-Wasserstein distance between the ground truth data distribution and the model's reconstruction distribution.

**Strengths:**

The authors' bounds are unique among supersample generalization bounds in that they explicitly involve the information gain between the train-test split mask $U$ and the latent variables of the generative model $J$.

I have skimmed the proofs in Appendix C, which seem correct.

**Weaknesses:**

As someone who does not work directly on supersample methods but has past experience with statistical learning theory, I find the significance of the authors' results for the ICLR community unclear.

This assessment is due to a combination of two factors: 1) on the theoretical side, the authors' results for bounding the generalization error involve practically incomputable quantities and only work under restrictive assumptions. In particular, I find the supersample setting quite artificial. 2) The authors' experiments appear to show that their bounds are extremely loose, as they are four orders of magnitude larger than the generalization error they are attempting to upper bound.

As such, I don't believe that the authors' results support the practical design principles they advocate for, such as "To improve the generalization and data generation capabilities, it is desirable to use a complex decoder..." on line 436. The issue with this statement is that given how loose the authors' bounds are, following this advice (in the context of the paper) is only useful to make the bounds less vacuous and is far from guaranteed to lead to practical benefits. To make such claims, the authors would have to perform more detailed experiments demonstrating that this advice leads to improvements.

In a similar vein, as the last sentence of the abstract, the authors claim, "Finally, we guarantee the 2-Wasserstein distance between the true data distribution and the generated data distribution, offering insights into the model's data generation capabilities." - in a similar manner as above, after reading the paper, I do not feel like the authors' results bear out this claim.

As things currently stand, I believe the paper is much better suited to a more specialized venue, such as COLT; this would also allow their claimed results to be scrutinized much more.

The paper's presentation could be improved a lot as well. The biggest missing component is providing intuition about the quantities and results; the authors should spend a lot more time explaining (to a machine learning audience) why the settings and results are sensible. I also suggest that the authors merge Section 1.1 with Section 3.4 and rewrite it in the same style, meaning that not only do they list the related works but also say how their own work is different. Wading through the literature review in the introduction is quite challenging and does not add much value to the paper.

**Questions:**

n/a

---

> ### Author Response · Authors · 2024-11-22
> **Author response (1)**
>
> We sincerely thank you for your feedback.
> Regarding the practical contributions of this study, we kindly refer you to our responses to the all reviewers for further details.
> Below, we address your other questions as follows.
>
> ## Q.1: Regarding the uncomputable and loose bound (Theorem 3), the assumptions, and the practicality of supersample
>
> ### Answer to concerns about the uncomputable and loose bound (Theorem 3):
> First, we would like to reiterate that the primary focus of this study is to provide a more realistic understanding of the generalization performance of VQ-VAE.
> As mentioned in our response to all reviewers, existing generalization error bounds are governed by the KL divergence over the parameter distributions of both the encoder and decoder, leaving the question of which component fundamentally contributes to generalization unresolved.
>
> Our bound presented in Theorem 2 addresses this question by being expressed in terms of the KL and the CMI term, evaluated from the latent variables obtained as the encoder’s output.
> This provides a clear answer: the encoder plays a fundamental role in generalization.
> Moreover, the bound in Theorem 2 is derived by extending existing supersample settings to produce numerically computable results; therefore, our bound is also numerically calculable.
>
> The bound in Theorem 2 is expressed in terms of two complexity measures: CMI and KL. This naturally raises the question of which measure correlates more strongly with the actual generalization error. Our numerical experiments were designed to address this question.
> The results confirmed that CMI exhibits a stronger correlation with the behavior of the generalization error compared to KL. We also observed that KL suffers from the drawback of failing to converge as n increases, as discussed in Eiger & Koch (2019) and Sefidgaran et al. (2023).
>
> This observation motivated us to explore an upper bound that excludes the problematic KL term from the bound in Theorem 2, leading to the findings presented in Theorem 3.
> Indeed, as you correctly pointed out, this result currently has the limitation of being numerically intractable. However, in the context of (VQ-)VAE, it represents the first theoretical bound that is asymptotically guaranteed to converge to $0$ as $n$ increases.
> By extending this result to a numerically computable bound in the future, we aim to provide a more practical and realistic generalization error bound. We believe that opening this possibility is another significant contribution of our study.
>
> Thanks to your insightful comments, we realized that reorganizing the flow of our analysis as described above would make our presentation clearer.
> Therefore, in the revised manuscript, we have moved the numerical experiments presented in Section 5 to Section 3 and reorganized Sections 3 and 4 according to the flow outlined above.
>
> (Continue to the next thread.)

---

> > ### Author Response · Authors · 2024-11-22
> > **Author response (2)**
> >
> > ### Answer to concerns about the assumptions and supersample settings:
> > The assumptions in this study are as follows: (1) boundedness of the instance space, (2) evaluation of reconstruction loss using the squared loss, and (3) modeling as a discrete latent variable model. Regarding assumption (3), this includes the class of VQ-VAEs, which are widely used in practice. However, extending the results to standard Gaussian latent variable models, such as VAEs, would require a fundamental change in the proof technique.
> > As for assumptions (1) and (2), which are related to the loss, we believe they are not particularly strong, as they are satisfied in practical datasets and are widely used when employing VQ-VAEs.
> > Finally, the supersample setting is merely a hypothetical framework introduced for generalization error analysis. It does not imply that actual models must be trained in this manner, so it is not to be considered an assumption.
> >
> > The assumptions made in this study can be summarized as follows: (1) boundedness of the instance space, (2) evaluation of the reconstruction loss using the squared loss, and (3) modeling as a discrete latent variable model. Assumptions (1) and (2) are conditions imposed on the loss function. Assumption (1) is a property that is typically satisfied when working with real-world datasets, and assumption (2) is mild, as the squared loss is widely used when applying VQ-VAE.
> > Assumption (3) holds for the class of VQ-VAE models analyzed in this study. However, as noted in your comment, this assumption does not hold for VAE models based on standard Gaussian latent variables. Consequently, our proof techniques cannot be directly applied to standard VAEs, and new analytical approaches would need to be developed. Although theoretical analysis of standard VAEs is out of the scope of this paper, we recognize it as an important direction for future work.
> > Finally, we would like to emphasize that the supersampling framework is not intended for use in the model's actual training process.
> > This framework is adopted to evaluate a model's generalization performance after training. Therefore, it should not be considered an assumption made during the training phase. For further details, please refer to the following papers.
> >
> > - H. Harutyunyan, M. Raginsky, G. Ver Steeg, and A. Galstyan. Information-theoretic generalizationbounds for black-box learning algorithms. In Advances in Neural Information Processing Systems, pp. 24670–24682, 2021.
> > - F. Hellström and G. Durisi. A new family of generalization bounds using samplewise evaluated CMI. In Advances in Neural Information Processing Systems, 2022.
> > - F. Futami and M. Fujisawa. Information-Theoretic Generalization Analysis for Expected Calibration Error. In Advances in Neural Information Processing Systems, 2024.
> >
> > ## Q.2 Regarding the statement of Theorem 5. (2-Wasserstein distance)
> > A. Theorem 5 was developed to explore how the error between generated images, transformed through the decoder from latent variables sampled from the prior distribution (i.e., the latent space), and the true image data distribution can be evaluated.
> > The results showed that, similar to our generalization error bound, this error can be evaluated based on the reconstruction error computed on the training data and the complexity of the encoder.
> > This finding theoretically guarantees that the encoder’s complexity is fundamentally involved not only in the reconstruction error of individual images but also in the error measured by the distance between data distributions. This fact further illustrates the critical role that the encoder plays in data generation.
> >
> > In this context, we used the term ‘theoretically guaranteed.’ We apologize for any confusion caused by our initial explanation and have revised the text around Theorem 5 to clarify the above points.

---

> > > ### Author Response · Authors · 2024-11-22
> > > **Author response (3)**
> > >
> > > ## Q.3 Regarding the relationship with prior work
> > > Based on your suggestion, we have reorganized the discussion as follows:
> > >
> > > As mentioned in the paragraph starting from line 384, the most relevant studies to our work are Mbacke et al. (2023) and Sefidgaran et al. (2023).
> > > Mbacke et al. (2023) analyzed the generalization error of VAEs using PAC-Bayes analysis with respect to reconstruction error. However, this analysis assumes that the decoder is fixed and independent of the training data, which makes it impractical.
> > >
> > > On the other hand, Sefidgaran et al. (2023) investigated the generalization performance of latent variable models in the context of classification tasks.
> > > In classification tasks, latent variables are learned through the encoder, and the decoder is used to output predicted labels.
> > > Their generalization error analysis shares commonalities with our work in two key aspects: (1) both the encoder and decoder are trained using the same training data, and (2) the generalization error bound depends solely on the encoder.
> > > However, their results are restricted to the classification tasks and cannot be used to analyze the reconstruction loss and data generation capability of the VAEs.
> > >
> > > In contrast, our study extends these findings under a more general image generation task in VAEs while maintaining the natural assumption in (1).
> > > We provide the theoretical insight described in (2) through an analysis based on the symmetric supersampling framework proposed in our work.

---

> > > > ### Author Response · Authors · 2024-11-30
> > > > **Any discussion?**
> > > >
> > > > Dear Reviewer FwXK,
> > > >
> > > > Thank you very much for taking the time to review our paper.
> > > >
> > > > We hope that our responses have addressed your concerns satisfactorily.
> > > > As the discussion period is nearing its end, we kindly ask you to let us know if there are any remaining points that require further discussion. If our responses have resolved your concerns, we would be grateful if you could consider revisiting your score.
> > > >
> > > > Thank you for your time and consideration.
> > > >
> > > > Sincerely,
> > > > --Authors

---

> > > > > ### Comment · Reviewer_FwXK · 2024-12-01
> > > > >
> > > > > I thank the authors for their rebuttal and apologise for the tardiness of my response.
> > > > >
> > > > > Unfortunately, the authors' rebuttal left my opinion unchanged.
> > > > >
> > > > > As the authors state:
> > > > > > the supersample setting is merely a hypothetical framework introduced for generalization error analysis. It does not imply that actual models must be trained in this manner, so it is not to be considered an assumption.
> > > > >
> > > > > That's fair enough, but it's quite difficult to see how the authors' results should translate to actual practical circumstances. This issue is compounded by the looseness of the numerical estimates for the authors' bounds, as I pointed out in my review.
> > > > >
> > > > > I should emphasize that I do think the authors have done valuable work; my concern is with its relevance to the ICLR community. However, should the AC/other reviewers believe the work is good enough to appear at the conference, I will not mind being overruled.

---

> > > > > > ### Author Response · Authors · 2024-12-01
> > > > > > **Reply from authors**
> > > > > >
> > > > > > We deeply appreciate your careful review of our responses and your recognition of the value of our work. It is truly an honor and a source of great happiness for us.
> > > > > > We are also pleased to hear that you found the supersample setting to be fair enough.
> > > > > >
> > > > > > ## Regarding “actual practical circumstances”
> > > > > > We would like to emphasize that the core contribution of our work lies in providing a more accurate theoretical understanding of the generalization performance of VQ-VAE, specifically aimed at facilitating “more practical evaluations of generalization performance.” Under the supersample setting, we have demonstrated that CMI is more suitable than KL for evaluating the generalization performance of VQ-VAE, as shown in prior studies. This insight led to the development of our alternative bound (Theorem 3), which aims to provide “more practical” generalization performance evaluations.
> > > > > >
> > > > > > As we acknowledge in the Limitation section, this evaluation remains challenging and leaves room for further improvement in achieving practical generalization performance evaluations. However, our work has already made a significant step forward by theoretically elucidating that the encoder plays a critical role in generalization, addressing the ambiguity in prior research about whether the encoder or decoder contributes more to generalization. This finding itself is highly beneficial for practitioners when designing their models.
> > > > > >
> > > > > > That said, there might be a discrepancy between our intended notion of “practicality” and what you envision as “actual practical circumstances.” If our response above does not fully address your concerns, we would greatly appreciate it if you could clarify your perspective on “actual practical circumstances” more concretely.
> > > > > >
> > > > > > ## Regarding the ICLR community’s interest in our study
> > > > > > First, as you already know, VAE was originally introduced at ICLR, and its extension, VQ-VAE, along with its generalization performance analysis, has been presented at NeurIPS, a conference with high relevance to ICLR ([Van Den Oord et al., NeurIPS2017; Mbacke et al., NeurIPS2023; Sefidgaran et al., NeurIPS2023]).
> > > > > >
> > > > > > Moreover, generalization error analysis, like the one we conducted, has also been actively discussed at ICLR ([Wang et al., ICLR2022; Dong et al., ICLR2024]), which demonstrates the ICLR community’s strong interest in both VAE and generalization error analysis.
> > > > > >
> > > > > > From these points, it is reasonable to conclude that our research aligns with the interests of the ICLR community.
> > > > > > (The references mentioned above are already cited in our manuscript.)
> > > > > >
> > > > > > We hope that our responses have adequately addressed your concerns. Thank you once again for your invaluable feedback and constructive discussion.

---

### Author Response · Authors · 2024-11-22
**Author response for all reviewers**

First and foremost, we would like to express our gratitude to the reviewers for considering our paper.
It appears that many of you share a common question regarding the practical insights this study offers to the ML community.
As a collective response to this concern, we would like to provide our explanation below.

### 1. Regarding the overall insights provided by the generalization analysis of latent variable models.

Given the increasing prominence of latent variable models, such as diffusion models, understanding the role that latent variables play in achieving generalization through theoretical analysis is expected to contribute to the development of more advanced latent variable models in the future.
To support this potential, we focused on the practical VQ-VAE and conducted this study.

### 2. Practical implications of our theoretical findings.

Our main theoretical results, Theorems 2 and 3, demonstrate that when reconstruction error is measured by the squared loss, the generalization results hold for “any decoder structure.”
This highlights that increasing the complexity of the decoder does not contribute to generalization performance. Instead, the generalization performance solely depends on the complexity of the encoder.
Furthermore, applying our theoretical results to a decoder different from the one used to train the latent variables still yields the same upper bound, further supporting the practical importance of our theoretical findings.

Additionally, our discovery aligns with existing studies. Many VAEs developed to date are trained by variational Bayesian principles or the information-bottleneck-based algorithms based on primarily regularizing the encoder rather than the decoder.
Our conclusion that encoder complexity is the key determinant of generalization performance is consistent with these prior studies.

The traditional generalization error bounds presented in line 205 and Appendix B depend on the complexity of both the encoder and the decoder.
As a result, it has been unclear whether high generalization performance is attributed to the decoder’s contribution or the encoder’s extraction of meaningful representations.
Previous results merely concluded that generalization performance is achieved through the combined contributions of both components, leaving the practical question of whether to prioritize encoder or decoder improvements unresolved.

Our new upper bound directly addresses this issue.
It explicitly shows that to improve the generalization performance of VQ-VAE, practitioners should focus solely on designing an appropriate encoder.
We believe this insight will benefit practitioners as they explore models to achieve higher generalization performance.

We have incorporated the above discussion into our revised manuscript (Appendix F.3).

---

### Meta-Review · Area_Chair_Mcak · 2024-12-19

**Metareview:**

**Summary of Discussion:**
The paper presents an information-theoretic generalization analysis for Vector-Quantized Variational Autoencoders (VQ-VAEs), aiming to understand the role of latent variables in generalization and data generation capabilities. While the theoretical work extends existing bounds to unsupervised settings, significant concerns were raised regarding the practical utility and novelty of the contributions.

**Key Concerns:**

1. **Looseness of the Bounds:**
   - The derived generalization bounds are several orders of magnitude larger than the actual generalization gaps observed in experiments, rendering them vacuous for practical purposes.
   - The bounds being four orders of magnitude off limits their applicability and fails to provide meaningful insights into model performance.

2. **Limited Novelty in Extension:**
   - Extending existing generalization bounds from supervised to unsupervised settings without addressing the inherent looseness does not constitute a substantial contribution.
   - Similar bounds and technical challenges have been previously studied in supervised contexts, and the paper does not offer significant advancements over these prior works.

3. **Lack of Practical Implications:**
   - The theoretical results do not translate into actionable guidance for improving VQ-VAE training or performance.
   - Experiments focus on investigating correlations between terms in the bounds and the generalization gap but do not demonstrate practical benefits or applications of the analysis.

4. **Clarity and Focus:**
   - The paper could better emphasize the technical reasons behind the looseness of the bounds and explore potential ways to tighten them.
   - Key assumptions and definitions require clearer presentation to enhance understanding and accessibility for the broader community.

**Conclusion:**
While the paper addresses an important topic in the theoretical understanding of unsupervised learning models, the current contributions are insufficient for acceptance. The significant gap between the theoretical bounds and practical observations limits the usefulness of the results. A more impactful contribution would involve developing tighter bounds or providing practical insights that can guide the design and training of VQ-VAEs.

**Additional Comments On Reviewer Discussion:**

See above

---

### Decision · Program_Chairs · 2025-01-22

Reject